# Using smartphones to optimise and scale-up the assessment of model-based planning

Kelly R. Donegan [1,2], Vanessa M. Brown[3], Rebecca B. Price[3], Eoghan Gallagher[1,2], Andrew Pringle[1,2], Anna K. Hanlon[1,2] & Claire M. Gillan [1,2,4✉]

Model-based planning is thought to protect against over-reliance on habits. It is reduced in individuals high in compulsivity, but effect sizes are small and may depend on subtle features of the tasks used to assess it. We developed a diamond-shooting smartphone game that measures model-based planning in an at-home setting, and varied the game's structure within and across participants to assess how it affects measurement reliability and validity with respect to previously established correlates of model-based planning, with a focus on compulsivity. Increasing the number of trials used to estimate model-based planning did remarkably little to affect the association with compulsivity, because the greatest signal was in earlier trials. Associations with compulsivity were higher when transition ratios were less deterministic and depending on the reward drift utilised. These findings suggest that model-based planning can be measured at home via an app, can be estimated in relatively few trials using certain design features, and can be optimised for sensitivity to compulsive symptoms in the general population.

[1] School of Psychology, Trinity College Dublin, Dublin, Ireland. [2] Trinity College Institute of Neuroscience, Trinity College Dublin, Dublin, Ireland. [3] Department of Psychiatry, University of Pittsburgh School of Medicine, Pittsburgh, USA. [4] Global Brain Health Institute, Trinity College Dublin, Dublin, Ireland. ✉email: gillancl@tcd.ie

Model-based or goal-directed planning is a cognitive capacity that involves building a mental map of potential action-outcome links and using that to make considered, flexible and optimal decisions[1,2]. A consistent finding in the literature suggests that compulsive behaviours, as seen in Obsessive-Compulsive Disorder (OCD), addiction and aspects of eating disorders, are associated with impairments in model-based planning. This has been shown in online general population samples where individuals vary on a spectrum of compulsivity[3], in clinical cohorts[4,5], and is suggested to have a developmental origin[6]. Mechanistically, theories suggest these deficits arise due to a failure to create accurate internal models of the world[7,8], which leaves patients vulnerable to getting stuck performing habits[9]. Although the finding is consistent, like many studies assessing the relationship between cognition and mental health symptoms, the effect size is small. To progress our understanding of if and how model-based planning causally relates to compulsivity and develop real-world clinical or public health applications, we need to rethink how we measure it, in whom, and in what setting. One option is to consider population approaches - studying small behavioural effects such as this in larger samples, in real-world settings, and where possible, repeatedly through time. Smartphone-science is a promising way to achieve this, though there are concerns that a departure from the experimental control of a lab environment, coupled with changes to core design features of cognitive tasks may come at the cost of validity, reliability, and data quality. Indeed the latter has been a source of considerable debate in the cognitive neuroscience literature.

In recent years, several studies have raised issues with how alterations to key parameters of a task commonly used to assess model-based planning, the two-step task, can affect its measurement. One of the earliest studies in this area illustrated that model-based planning is reduced when there are concurrent working memory demands, and that this reduction depends on individual differences in working memory capacity[10]. Kool and colleagues[11] gathered data on two versions of a two-step task; the original version developed by Daw et al.,[1] and a modified version, which their simulations suggested would increase the incentive value of engaging in model-based planning. They found that the modified version (which included several changes to key task parameters) indeed elicited greater model-based planning compared to the original. Others have shown through simulation that changes to reward probabilities may undermine the validity of standard analyses of the task[12]. For example, in cases where reward probabilities are unequal (i.e., one-second stage state is more rewarding than another), simulated model-free agents can produce behaviour that appears model-based. In another study, they found model-based estimates were significantly greater when participants received in-depth instruction and more practice trials compared to the original task[13]. More recently, researchers assessed the emergence of model-based planning in a task that was initially absent of any instruction whatsoever[14]. They found that only a minority of participants adopted a model-based approach to solving a two-step task without instruction, and once instructions were provided, model-based planning estimates rose rapidly. Across all of these studies, an important facet remains untested – do these task variations shift model-based planning scores equivalently across individuals, or do alterations to task design fundamentally change the meaning of the quantity under study, i.e., its external validity. Recent work suggests mixed evidence. Assessing differences in task motivations, Patzelt et al.,[15] found that offering larger reward amounts increased mean-level model-based planning levels, but this did not affect its association with compulsivity. Castro-Rodrigues[14], on the other hand, found evidence to suggest that differences between OCD patients and controls may be smaller when detailed instructions are provided,

though the sample ($N = 46$ OCD patients) was perhaps too small to test this definitively.

A second and important issue is how task modifications affect reliability, which sets a ceiling for the size of the association one can observe with compulsivity. Test-retest estimates of model-based planning from the traditional task have been mixed, ranging from poor to good (r = [0.14–0.40])[16], or non-existent to excellent (r = [−0.10–0.91], median = 0.45)[17], depending on analytic choices. This finding is common to many tasks used for individual difference research[18] and is thought to in part be the result of the reliability paradox, where tasks designed to examine within-subject effects (such as Flanker, Stroop) have low between-subject variability[19]. One simple way to increase reliability is to increase the amount of data (i.e., trial number) gathered per-participant[20]. While this helps, reliabilities eventually plateau, often below an acceptable level. For example, Stroop reaction time can become more reliable with additional trials to a point but intraclass correlation coefficient (ICC) values plateau around 0.4[19]. Similarly, Price et al.,[21] found relatively consistent ICC values for an attentional bias metric measured from just 48 trials compared to an estimation from 320 trials suggesting that the benefit of adding trials may be tenuous. Further, it is unclear if and how reinforcement learning tasks like the two-step task benefit from additional trials and importantly if improvements in reliability translate into improvements in external validity. For example, early trials might measure something qualitatively different from later trials, particularly in high-order cognitive tests where rules are learned and then deployed, allowing more automatic forms of behaviour to take over.

The present study aimed to address these issues by gamifying and then optimising a commonly-used task that tracks individual differences in compulsivity, testing if key features of task design and trial number could boost its reliability and external validity. This requires large samples, and so we developed a diamond-shooting game called Cannon Blast that could be played by members of the public, aka citizen scientists[22], from anywhere in the world in an at-home environment. Cannon Blast was designed to be fun and repeatable, but critically it contained key features of the classic two-step task allowing us to assess model-based planning. We aimed to validate the game in two ways: first by establishing that it elicits model-based behaviour similar to the traditional task and then by demonstrating that model-based estimates correlate across tasks. Next, we released the game to the general public through our labs non-profit Neureka app (http://www.neureka.ie), and by leveraging large-scale data collection, aimed to test if the estimates of model-based planning derived from the gamified task would show the same associations with demographic individual difference measures such as older age[23,24], female gender[3] and lower IQ and processing speed[3,25] but also specific negative associations with compulsivity[3,5,26]. Finally, we wanted to utilize these associations as 'ground truth' to assess if the external validity of model-based planning estimates are affected by modifications to the task set-up. We compared transition probabilities that were more or less deterministic (80:20 vs. 70:30), used different sets of drifting reward probabilities, varied concurrent task demands (i.e., the difficulty of the diamond shooting task itself), compared earlier vs later trials of the game and tested the impact of increasing trial numbers.

## Methods

The procedure and statistical plan for both experiments described below were not preregistered.

**Ethical considerations and data protection.** This research was granted ethical approval by the Research Ethics Committee of the

School of Psychology at Trinity College Dublin (Approval number: SPREC072019-01). The Neureka app is a non-profit smartphone application developed and maintained by the Gillan Lab, Trinity College Dublin. For Experiment 1, prospective participants received an information sheet and gave informed consent through the online survey platform Qualtrics. Participants across both experiments were required to also read the information sheet and consent to participation embedded to the registration process for Neureka. This described the wider scientific aims of the Neureka Project, what participation involves, terms of data use, data protection procedures, health risks, withdrawal of data procedures and points of contact. For more detail on the exact contexts of this information sheet provided to participants, see Supplementary Note 1. Data collected through Neureka is stored and processed in accordance to EU General Data Protection Regulations.

## Experiment 1

*Participants.* We recruited participants to complete the traditional two-step task in a web-browser and Cannon Blast in the smartphone app Neureka. We targeted a minimum sample size of $N = 50$, which provides 80% power to detect a medium effect with a significance level set at $p < 0.05$. To allow for data-loss and exclusions, data were collected from $N = 68$ participants who were 18 years or older and have access to both smartphone and computer devices with an internet connection. Participants were compensated €10 upon completion of both tasks. Post exclusion criteria, $N = 57$ remained for analysis (43 women (66%) and 14 men (34%) aged between 18–46 ($M = 22.95$, SD $= 5.6$)). Gender-identification was collected in-app by asking 'What gender do you most identify with" and a list of seven options: male (hereafter 'man'), female (hereafter 'woman'), transgender male (hereafter 'non-cisgender'), transgender female (hereafter 'non-cisgender'), non-binary (hereafter 'non-cisgender'), not-listed (hereafter 'non-cisgender'), or prefer not to say.

*Procedure.* Participants were recruited and tested online. During the sign-up process, they provided electronic consent, along with self-reporting basic demographic (age, gender, education) and eligibility information. They completed the traditional two-step task in a web-browser on a laptop or desktop computer and Cannon Blast on an iOS or Android smartphone. The order of task was counterbalanced across subjects and the entire study took less than 60 min.

Cannon Blast: The goal of Cannon Blast is to hit as many diamonds as possible in 100 shots (Fig. 1a). On each trial, participants first aimed their cannon and then selected between two containers containing purple and pink balls. The left cannon always contained more purple balls (80%) and the right more pink balls (80%). In contrast to the traditional task, this transition structure did not have to be learned or remembered; it was visibly displayed on-screen i.e., each container possessed eight balls of the corresponding colour and two balls of the alternate colour. After the container was selected, a ball was randomly pulled from this container, and was consistent with the most prominent colour (80%, a common transition) or produced the minority colour (20%, a rare transition) (Fig. 1b).

On what we define as rewarding trials, participants received a good ball that exited from the cannon and could be used to hit the diamond (Fig. 1c). There was no guarantee that a good ball would actually hit a diamond, this depended on the participant's aim and timing. Alternatively, participants were unrewarded if the ball disintegrated upon firing, thus reducing the chance of hitting a diamond to zero. The probability of being rewarded (in other words receiving a good ball) drifted independently over the course of the task, much like the second-stage outcomes in the traditional task. However, the traditional task typically utilises a single pre-determined drifting reward probability structure for the 200 trials of the task. To allow us to assess the potential impact of drift dynamics on parameter estimation (in Experiment 2), Cannon Blast instead used two possible drift structures for each block of trials. Participants were randomly assigned drift A or drift B for each of their 100 trial blocks, leading to a total of four reward probability drift sets combinations for Cannon Blast participants (A-A, A-B, B-A, B-B: Fig. 1d).

The similarities and differences between the original two-step task and our gamified Cannon Blast are presented in Supplementary Table 1. Like the original, Cannon Blast consisted of 200 trials which was divided into two blocks of 100 trials. The first block was set at an easy difficulty, and the second at a medium difficulty. Level had no direct bearing on the core parameters-of-interest (which container participants select, rewards, drifts etc), and instead reflected how challenging the aim and shoot trajectory was. However as we explore in Experiment 2, level difficulty can be conceived of as a distraction manipulation. Easy levels included trials where the diamond did not move, had static obstructions that limited the angle at which it could be hit, or where diamonds moved slowly around the screen. Medium difficulty levels included more challenging trials with both moving diamonds and moving obstructions (Supplementary Table 2). While on average medium trials are more difficult than easy (average hit rate Medium=45%, Easy=52%), there was variation within both Easy levels (hit rates 83%, 53%, 44%, 29%) and Medium (hit rates 75%, 20%, 39%, 45%) (Supplementary Table 2). Reward probabilities in Cannon Blast were set higher on average (average reward probability: AA $= 0.80$ [0.64–0.94], AB $= 0.72$ [0.41–0.94], BA $= 0.72$ [0.41–0.94], BB $= 0.64$ [0.41–0.94]) than the original task (mean reward probability $= 0.52$ [0.25–0.75]) to promote enjoyment and limit frustration. In contrast to the traditional two-step task, which includes 40 practice trials, Cannon Blast starts with a short, passive walk-through demonstration of the task (Supplementary Figure 1). While the traditional task design has both first- (choice of rocket) and second-stage (choice of alien) actions, Cannon Blast has first-stage (choice of container) actions only. The decision to remove second-stage actions was in part done for gameplay reasons but also has been shown to increase the importance of model-based contributions in the first stage choice[11]. A final major distinction between the tasks was the stated goal; in the traditional task, participants are directly told to earn rewards (space treasure). In Cannon Blast, participants are told to shoot as many diamonds as possible, and that this can be facilitated by ensuring they maximise rewards (good balls).

Traditional two-step reinforcement learning task: Participants completed an adapted version of the two-step reinforcement learning task[1], developed by Decker et al.[11]. The contents of this task have been described in detail in Decker, et al.[27] and are summarised Supplementary Figure 2 and Supplementary Table 1.

## Data analysis

*Exclusion criteria.* Participants were excluded from the traditional task if they: (a) missed more than 20% of trials ($N = 2$)[7], (b) responded with the same key press at the first stage of the task on more than 95% of trials ($N = 5$)[3]. Exclusion criteria for Cannon Blast were harmonised with these as much as possible. We excluded participants if they had (a) missed more than 20% of trials ($N = 1$) or (b) selected the same container more than 95% of the time ($N = 4$). However, it is important to note that for

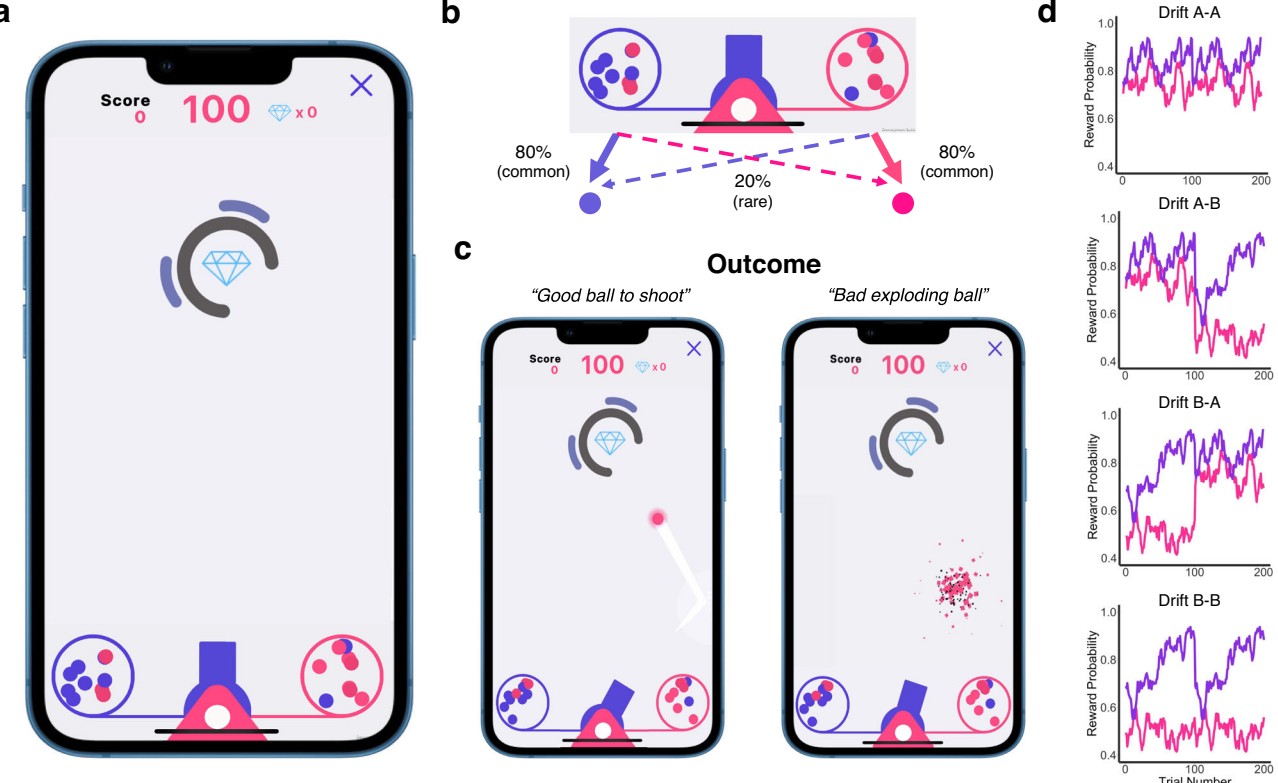

**Fig. 1 Task structure of Cannon Blast, a smartphone game to assess model-based planning. a** In this game, participants' goal is to shoot as many diamonds as possible before their total number of shots (100 per block) runs out. To do so, they must aim a central cannon and then select which circular container to draw from. **b** Purple and pink balls dynamically bounce around each of the flanked containers which depict the probability of a pink or purple ball being released. For example, the left container displays 8 purple balls and releases a purple ball 80% of the time (a common transition) and displays 2 pink balls, giving a pink ball on 20% of trials (a rare transition). **c** The purple and pink balls have different values that dynamically change throughout the game. The value of the ball is defined as the probability of it being a 'good ball', i.e., one that remains intact after firing (rewarding trial), or a dud ball (non-rewarding trial) that explodes shortly after being fired, and therefore cannot reach the diamond. **d** We included 2 drifting reward probabilities (A, B) that quantively differed on various metrics (see Supplementary Table 6). Participants were randomly assigned a reward drift set at each block leading to four distinct drift set combinations (A-A, A-B, B-A, B-B).

criterion (a) as there was no time limit to make this response (unlike the traditional task), participants could not miss trials due to being too slow or disengaged (unless they quit the experiment entirely). Notwithstanding, we noted that some trials were missing for 2 users from our app database (presumably due to a technical glitch) and for one of these, this exceeded the 20% threshold and were therefore excluded. Combining all exclusion criteria for both tasks, $N = 11$ (16%) participants were excluded with $N = 57$ remaining for analysis (37 women, aged between 18–32 ($M = 22.01$, SD $= 4.12$)).

*Quantifying model-based planning.* All analysis were performed through RStudio version 1.4.1106 (http://cran.us.r-project.org). Across both task versions, data distribution was assumed to be normal (Fig. 2a, b) but this was not formally tested. Hierarchical logistic regression (HLR) models, which are mixed effects models for a binary outcome variable, were conducted using mixed effects models implemented with the lme4 package in R. The model tested if participants' choice behaviour in the first stage state (coded as switch: 0 and stay: 1, relative to their previous choice) was influenced by reward (coded as unrewarded: $-1$ and rewarded:1), transition (coded as rare: $-1$ and common: 1), and their interaction, on the trial preceding. Within participants factors (main effect of reward, transition and their interaction) were modelled as random effects. Model-based index (MBI) is quantified as the interaction between Reward (traditional task:

space treasure vs. dust; Cannon Blast: good vs. dud ball) and Transition (traditional task: common vs rare transition to a planet from the chosen rocket; Cannon Blast: common vs rare ball colour appearing from the chosen container). In line with prior work on the traditional task, we also quantified model-free index (MFI: the main effect of Reward) and choice repetition (the Intercept of the model). Individual estimates for each parameter (MBI, MFI, choice repetition (hereafter 'stay') and transition) were extracted for each task and compared across tasks using Pearson correlation. We assessed the internal consistency of each task using split-half correlation (odds-even split method) using the guidelines from Cicchetti[28]: <0.4 poor, 0.4–0.7 fair, 0.7–0.9 good and >0.9 excellent.

## Experiment 2
*Participants.* Between June 2020 and October 2022, we collected data from 7466 unpaid Citizen Scientist users of the Neureka app. After applying exclusions detailed below, $N = 5005$ remained for analysis with 3225 (64%) women, 1683 (34%) men, 82 (2%) who did not identify as cisgender (including transgender, non-binary or not listed) and 15 who preferred not to disclose (0.3%) (Supplementary Table 3). The sample included a wide age range of 18–84 ($M = 45.38$, SD $= 14.54$) and 64% had attained post-secondary degree or higher ($N = 3220$). The sample was well-powered for individual difference analysis; a priori power analysis based on a prior paper[3] indicated that a sample size of $N = 541$

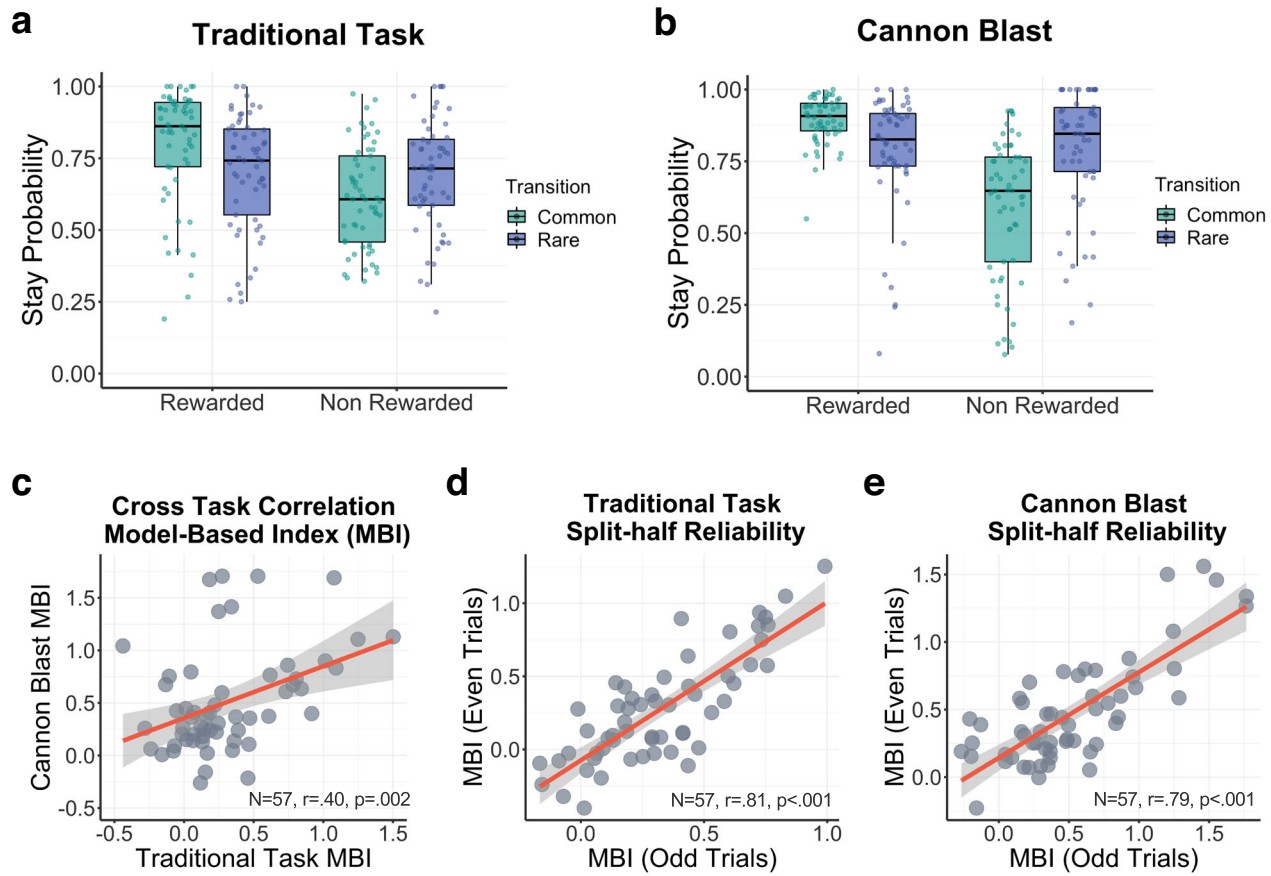

**Fig. 2 Validation of Cannon Blast against the traditional two-step task (N = 57).** Box-plot of stay probabilities across all trials and all participants from (**a**) the tradition task and (**b**) Cannon Blast. Each dot represents a participant. **c** Model-based indices (MBI) from the traditional two-step task positively correlated with model-based indices derived from Cannon Blast. Internal consistency of model-based indices in traditional task (**d**) and Cannon Blast (**e**) using the split-half odds-even reliability approach. For **c**–**e**, each dot represents a participant, the red line indicates line of best fit while grey area represents 95% confidence interval.

participants was required to achieve 80% power (two-tailed test, $p < 0.05$) in detecting the association between self-reported compulsivity symptoms and model-based planning measured from the traditional task using an online paid sample, controlling for covariates of age, gender and IQ ($r = 0.12$).

*Procedure*
Cannon Blast: Data was collected from citizen scientists who downloaded the Neureka app from the Play Store (Android users) or Apple's App Store (iPhone users), provided informed consent and completed tasks designed to help scientists learn about the brain. Participants completed Cannon Blast within the Neureka app in two ways: as part of a science challenge called Risk Factors designed to measure aspects of cognition and individual-level risk factors, or as a stand-alone challenge within the Free Play section of the app. Risk Factors included two other games along with a battery of self-report sociodemographic and lifestyle questionnaires which were used to collect information on participants age, gender and education. The order in which games and questionnaires were presented where pseudo-randomised such that a game was always delivered first, followed by alternating blocks of questionnaires and games. The majority of users completed the version of Cannon Blast described in Experiment 1 ($N = 2884$), but $N = 2138$ (gathered from July 2021 onwards) completed a version where the transition structure was set to 70:30, allowing us to examine the impact of transition probability on model-based planning. In Free Play participants could

reengage with Cannon Blast as a stand-alone game. Here, participants selected a difficulty level of their choice and completed 100 trials in that setting. This allowed us to examine the test-retest reliability of model-based planning, assess how collecting more trials per participant affects the reliability and validity of estimates, and disentangle the impact of block difficulty (Easy, Medium) from order effects ($1^{st}$ Block, $2^{nd}$ Block). An additional Hard difficulty level was available in the Free Play section only, with even more challenging diamond movements and obstacles to navigate.

Self-report psychiatric questionnaires and transdiagnostic factors: The Neureka app contains another science challenge called My Mental Health, where participants can complete validated questionnaires assessing nine aspects of mental health (209 items). Of the $N = 5005$ participants for whom we had Cannon Blast data, $N = 1451$ additionally completed this section. The nine questionnaires were used to measure alcohol dependency (Alcohol Use Disorder Identification Test, AUDIT:[29]), apathy (Apathy Evaluation Scale, AES:[30]), depression (Self-rated Depression Scale, SDS:[31]), eating disorders (Eating Attitudes Test, EAT-26:[32]), impulsivity (Barratt Impulsivity Scale, BIS-11:[33]), obsessive compulsive disorder (Obsessive Compulsive Inventory Revised, OCI-R:[34]), schizotypy (Short Scale for Measuring Schizotypy, SCZ:[35]), social anxiety (Liebowitz Social Anxiety Scale, LSAS:[36]) and trait anxiety (Trait portion of the State-Trait Anxiety Inventory (STAI:[37]. All questionnaires were presented to

participants in a randomized order. Mean and standard deviations of the mental health questionnaire total scores along with their associations with age, gender and education are presented in Supplementary Table 4. Internal consistency of all questionnaires ranged from good to excellent, Cronbach's α = [0.87–0.95]. Prior work has shown that these item level responses can be summarised as three transdiagnostic factors Anxious-Depression (AD), Compulsivity and Intrusive Thought (CIT), and Social Withdrawal (SW)[3]. We applied the weights from Gillan et al.[3] to the Neureka data for all analyses reported here, but additionally repeated the factor analysis to ensure that there were no major differences across the samples. Correlations between the weights derived from Gillan, et al.[3] and the present study were very high for each dimension, r = [0.95–0.97]. Noteworthy, there was a typographical error in the response options. Sensitivity analysis presented in the online supplement suggest this has little bearing on the results, and indeed the correlation across derived factors from the two datasets suggests a high degree of consistency (Supplementary Method 2). Other science challenges within the app collected data to derive transdiagnostic scores of compulsivity and anxious-depression using a reduced set of items from those scales. Previous work has validated this reduced set against the original set of items[38]. Using these items, we had task and compulsivity data from $N = 2369$ participants. We used data from $N = 1451$ who had compulsivity score from the full set of items to test for independent clinical associations with MBI along with testing a covariate model with anxious-depression, compulsivity and social withdrawal. We then used data of $N = 2369$ with a compulsivity score from reduced items in the analyses related to associating with game play metrics and also for testing the impact of task-optimizations on external validation.

### Data analysis

*Exclusion criteria.* As in Experiment 1, participants were excluded for (a) missing more than 20% trials on their first session ($N = 2394$, most of whom started, but did not complete the game), and (b) selecting the same container on more than 95% of the trials ($N = 797$). A further $N = 48$ were excluded for having incomplete demographic data required for external validation leaving $N = 5005$ remaining for analysis (33% data loss).

*Quantifying model-based behaviour.* We used the same basic HLR as in Experiment 1 for all analyses. Data distribution was assumed to be normal (Fig. 3a) but this was not formally tested. Additional analyses comparing the reliability and external validity of alternative approaches of deriving model-based planning scores as point estimates (PE) and using Hierarchical Bayesian modelling (HB) can be found in the supplementary material (Supplementary Method 3, Supplementary Table 18). Overall the results were highly similar, but there was an advantage for the HLR and so this estimation method was brought forward for analyses (Supplementary Method 4, Supplementary Table 19). Participant's individual regression co-efficient for the interaction between Reward and Transition (the model-based index, MBI) were extracted from the basic HLR and brought forward for analyses (e.g., to assess association with clinical and individual differences, test the impact of task-modifications and to assess split and test-retest reliability).

*Examining how task parameters affect model-based estimates.* We carried out a series of analyses designed to test if alterations to task parameters affect (i) mean MBI levels, (ii) its external validity: defined as the associations between MBI and individual difference measures of compulsivity, age, gender and education and (iii) it's internal consistency using split-half correlation and/

or test-retest reliability using intraclass coefficients. The structure of these analyses varied across parameter manipulations due to between vs within-subject manipulations and data availability. First, we examined transition structure, which was manipulated between-subject. As fewer participants experienced the 70:30 ($N = 2138$) transition ratio compared to the 80:20 ($N = 2884$), we down-sampled the 80:20 group and propensity score matched them on age, gender and education using the MatchIt R package (for descriptive information on this matching see Supplementary Table 3). Secondly, we manipulated task difficulty and order within-subject. We tested for differences in model-based planning estimates during the Easy (1st Block) vs. Medium (2nd Block) trials, and complemented this with analysis of Free Play data. As each block consisted of just 100 trials, we used the Spearman-Brown prophecy formula to assess reliability which allows for corrections when trial number is reduced[16,39]: corrected reliability = [2*reliability] / [1+reliability]. A subset of participants ($N = 785$) had repeated plays of Cannon Blast, accessed in the Free Play section of the app. These data allow us to disentangle practice/order effects from difficulty. Descriptive information relating to these Free Play sessions are presented in Supplementary Table 5. Thirdly, the classic version of this task includes a single drift sequence that all subjects experience and little is known about the implications of using different sequences. To address this, we randomised participants (i.e., between subject) to possible drifting reward probability conditions that differed in several potentially important dimensions, including their distinguishability, average reward rate and the changeability of reward probabilities (Supplementary Table 6). To avoid a wash-out of effects we tested the impact of drifts experienced in the first block of their first play. Also, to keep data as homogenous as possible, we decided to implement in a participants first play, only two out of the possible ten reward drifts used in app. In a between-subjects design, participants were either assigned Drift A ($N = 2139$) or Drift B ($N = 2345$). In repeated plays, participants were randomly assigned to one of the ten possible drifts (Drift A-J, Supplementary Figure 3). Unfortunately at the time of submission we did not have the sufficient power to present work using the other eight possible drifts. Finally, we tested the impact of trial number on model-based estimates. To do this, we examined within-subjects data from those who played at least 300 trials of Cannon Blast ($N = 716$). Here, we estimated MBI with varying trials collected per participant starting at 25 trials and increasing in bins of 25 trials until 300 trials.

**Reporting summary**. Further information on research design is available in the Nature Portfolio Reporting Summary linked to this article.

### Results

**Construct validity and reliability of Cannon Blast**. Cannon Blast embeds the classic two-step task structure (i.e., drifting rewards, a probabilistic transition structure) within a diamond shooting game. In this game, users aim a cannon at a diamond presented on screen, which might be static, moving around the screen or partially obstructed, depending on the difficulty level. Next, they select which of two containers they want to draw a ball from. The containers each have a mix of purple and pink balls; one has 80% pink balls and the other 80% purple, corresponding directly to the probability that a ball of that colour will be released. Not all balls work; some explode upon being released from the cannon. This is partially predictable from the colour of the ball, whereby the chances that a pink/purple ball will explode drifts slowly and independently over the course of the task. Unlike the traditional task, where rewards are an end onto

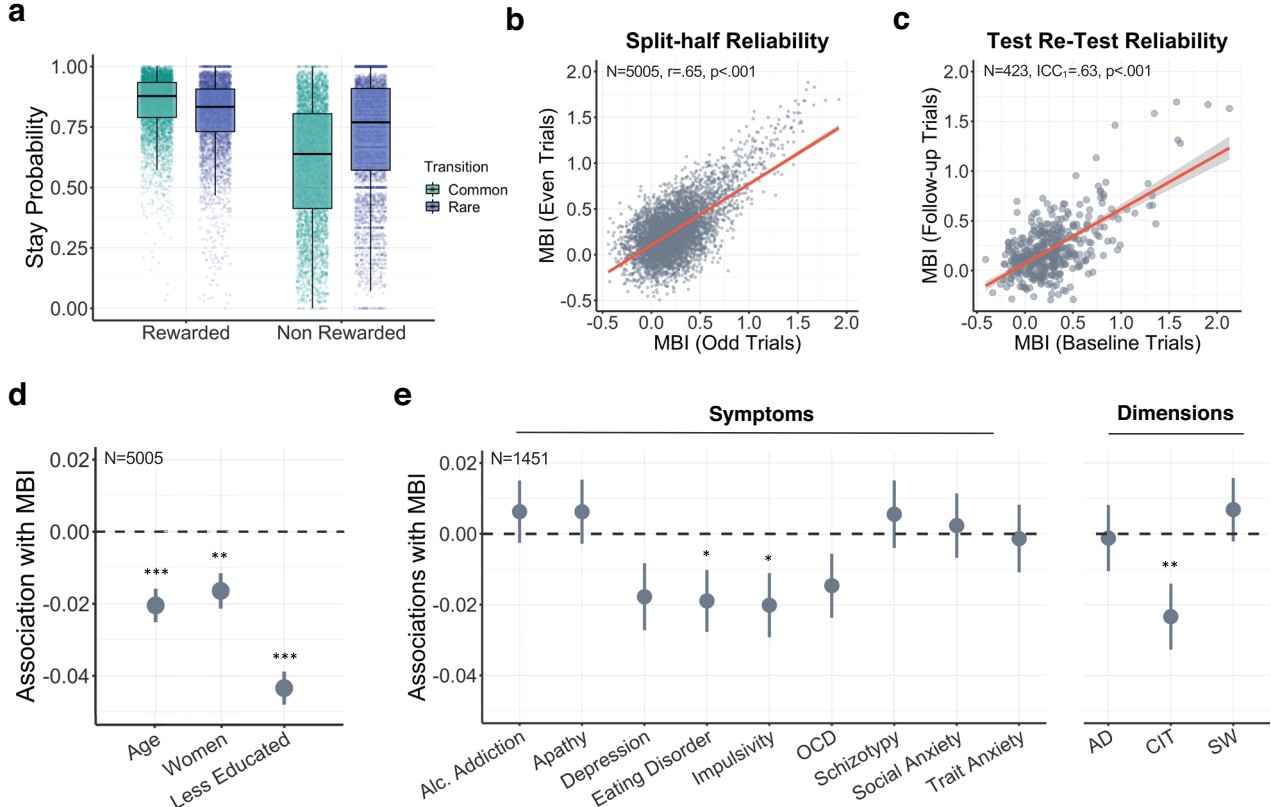

**Fig. 3 Large-scale external validation of Cannon Blast in citizen scientists (*N* = 5005). a** Box-plot of stay probabilities across all trials and all participants for Cannon Blast (*N* = 5005). Each dot represents a participant. **b**. Internal consistency of model-based index (MBI) using split-half (odds vs evens) method. **c**. Test-retest reliability of MBI from *N* = 423 who had 400 trials of Cannon Blast using intraclass correlation coefficient (ICC). MBI estimates from participants first 200 trials (Baseline Trials) plotted against MBI estimates from participants next 200 trials (Follow-up Trials). For **b**, **c**, each dot represents a participant, the red line indicates line of best fit while the grey area represents 95% confidence interval. **d**. Model-based associations with individual differences (age, gender, and education) in *N* = 5005 citizen scientists. Reductions in MBI associated with older adults, women and those less educated. **E**. Model-based associations with individual clinical questionnaires and transdiagnostic dimensions in a sub-sample (*N* = 1451) who had completed a battery of self-report clinical questionnaires. Greater levels of eating disorder and impulsivity symptom severity were associated with deficits in MBI. A transdiagnostic dimension of Compulsivity and Intrusive Thought (CIT) showed a specific association relative to Anxious-Depression (AD) and Social Withdrawal (SW). For **d**, **e**, error bars reflect the standard errors of mean. Both analyses controlled for age, gender, and education. *$p < 0.05$; **$p < 0.01$; $p < 0.001$***.

themselves, in Cannon Blast, rewards (i.e., a good ball to shoot with) have value insofar as they allow the user to shoot the diamond. This means there are two potential forms of reward in this task – getting a good ball and hitting the diamond. For clarity, we define Reward in Cannon Blast as the former, but unpack the impact of the latter on choice in our later analyses. To validate the task, 57 paid participants played Cannon Blast and the Traditional version of the task. Both tasks demonstrated a significant main effect of Reward (model-free index, hereafter 'MFI') and a Reward x Transition interaction (model-based index, hereafter 'MBI') (Table 1, Fig. 2a). In both tasks, participants tended to repeat choices across trials (i.e., a positive intercept), but the tasks differed in the main effect of transition, which was positive and negative in the traditional and gamified tasks respectively. To directly compare behaviour across the two tasks, an analysis of the entire dataset was conducted with Task Type (Cannon Blast, Traditional) as a fixed effect. This revealed a number of differences across tasks. Participants playing Cannon Blast tended to repeat their choices more often, they were more model-based, and less likely to stay following a common transition (Table 1). There was a moderate positive association between MBI derived from Cannon Blast and the Traditional task (r(55) = 0.40, p = 0.002, 95% CI = [0.16, 0.60], Fig. 2b). Split-half reliability for MBI were similar for both the traditional (r(55) = 0.81, p < 0.001, 95%

CI = [0.70, 0.88]) and Cannon Blast (r(55) = 0.78, p < 0.001, 95% CI = [0.66, 0.87], Fig. 2c).

Next, we replicated the findings reported above in a larger cohort of unpaid citizen scientists (*N* = 5005) who played Cannon Blast after downloading the smartphone app Neureka. There was evidence of model-free behaviour (main effect of Reward: β = 0.47, 95% CI = [0.45, 0.48], SE = 0.01, p < 0.001), model-based behaviour (Reward x Transition: β = 0.27, 95% CI = [0.25, 0.28], SE = 0.01, p < 0.001), an overall tendency to repeat choices (Intercept: β = 1.33, 95% CI = [1.30, 1.35], SE = 0.01, p < 0.001), and participants were more likely to stay following a rare transition (Transition: β = −0.08, 95% CI = [−0.07, −0.08], SE = 0.00, p < 0.001) (Fig. 3a; Supplementary Table 7). Internal consistency of the MBI was lower than in the smaller paid cohort (r(5003) = 0.65, p < 0.001, 95% CI = [0.63, 0.67], Fig. 3b). An intra-class correlation (ICC) was used to assess the test-retest reliability of the MBI from those who had played at least 4 blocks of Cannon Blast (i.e., 400 trials, *N* = 423) within a 30-day timeframe. Specifically, we compared MBI from their first session of Cannon Blast (200 trials, which were completed alongside questionnaires and other games) with their next 200 trials (completed in a section of the app where participants could repeatedly play Cannon Blast exclusively). We found test-retest reliability estimates were moderately associated (ICC$_1$:

**Table 1 Mixed effects logistic regression analysis of the Traditional Task and Cannon Blast.**

|  | β (SE) | z value | *p*-value |
|---|---|---|---|
| **Traditional Task** |  |  |  |
| Intercept | 1.11 (0.13) | 8.84 | <0.001*** |
| Reward | 0.35 (0.05) | 7.08 | <0.001*** |
| Transition | 0.11 (0.04) | 2.58 | 0.010** |
| Reward: Transition | 0.32 (0.07) | 4.82 | <0.001*** |
| **Cannon Blast** |  |  |  |
| Intercept | 1.51 (0.11) | 13.31 | <0.001*** |
| Reward | 0.48 (0.07) | 6.61 | <0.001*** |
| Transition | −0.11 (0.05) | −2.19 | 0.030* |
| Reward: Transition | 0.52 (0.08) | 6.13 | <0.001*** |
| **Full Comparison Model** |  |  |  |
| Intercept | 1.32 (0.09) | 15.31 | <0.001*** |
| Reward | 0.41 (0.05) | 8.86 | <0.001*** |
| Transition | −0.01 (0.04) | −0.16 | 0.873 |
| Task | −0.21 (0.09) | −2.46 | 0.014* |
| Reward: Transition | 0.42 (0.06) | 6.64 | <0.001*** |
| Reward: Task | −0.06 (0.04) | −1.48 | 0.142 |
| Transition: Task | 0.11 (0.03) | 3.98 | <0.001*** |
| Reward: Transition: Task | −0.10 (0.04) | −2.43 | 0.015* |

$N = 57$.
*$p < 0.05$, **$p < 0.01$, ***$p < 0.001$.
SE Standard Error.
Dependent variable in the model 'Stay' coded as (1,0: Stayed, Switched).
Independent variables in the model coded as: Reward (1,-1: Rewarded, Non-Rewarded), Transition (1,-1: Common, Rare); Task (1,-1: Traditional, Cannon Blast).

$r(421) = 0.63$, $p < 0.001$, 95% CI = [0.57, 0.67], Fig. 3c). Mean time elapsed between first and last session was 6.12 (±6.93) days, with a median interval of 4 days.

**External validation of Cannon Blast.** Using data collected from citizen scientists ($N = 5005$) playing Cannon Blast remotely, we replicated previous lab-based associations between MBI and individual differences (Fig. 3d, Supplementary Table 8). Older adults (β = −0.02, 95% CI = [−0.03, −0.01], SE = 0.00, $p < 0.001$), women (β = −0.02, 95% CI = [−0.03, −0.01], SE = 0.00, $p = 0.001$), and those with less education (β = −0.04, 95% CI = [−0.05, −0.03], SE = 0.00, $p < 0.001$) all showed reductions in MBI. $N = 1451$ completed an additional section of the app that contained a comprehensive battery of self-report mental health questionnaires. We observed associations between MBI and eating disorder symptoms (β = −0.02, 95% CI = [−0.04, −0.00], SE = 0.01, $p = 0.031$) and impulsivity (β = −0.02, 95% CI = [−0.04, −0.00], SE = 0.01, $p = 0.026$), controlling for age, gender, and education (Fig. 3e, Supplementary Table 8). We refactored the raw questionnaire items (209 items) into three transdiagnostic dimensions and defined them as; Anxious-Depression (AD), Compulsivity and Intrusive Thought (CIT) and Social Withdrawal (SW) based off previous work[3]. These factors were entered together as IVs in a model predicting MBI with age, gender and education controlled for. Consistent with prior work using the traditional task, we found a specific pattern of a statistically significant association between MBI and CIT (β = −0.03, 95% CI = [−0.05, −0.01], SE = 0.01, $p = 0.004$) but no statistically significant evidence for the association between MBI and AD (β = −0.00, 95% CI = [−0.03, 0.02], SE = 0.01, $p = 0.733$) or SW (β = 0.02, 95% CI = [−0.00, 0.04], SE = 0.01, $p = 0.106$) (Fig. 3e, Supplementary Table 8). For completeness, we also assessed the association of these individual difference measures and model-free and stay behaviour. There was no statisti-

cally significant evidence for the associations between individual differences or clinical associations with the tendency to repeat choices (Supplementary Table 9). However, individual differences in model-free learning behaved similarly to model-based planning in our task. The MFI was statistically associated with age, gender, education, and compulsivity, in the same direction as MBI (Supplementary Method 1; Supplementary Table 9).

**Broader patterns of gameplay by citizen scientists.** On average participants received rewarding good balls on 145/200 trials (~75%). Consistent with the set-up of the task, the more participants utilised a model-based approach, the more good balls they received (β = 1.76, 95% CI = [1.21, 2.31], SE = 0.28, $p < 0.001$); there was no statistically significant evidence for individual differences in model-free learning (β = 0.27, 95% CI = [−0.10, 0.63], SE = 0.19, $p = 0.149$). Those with less education were less likely to receive good balls (β = −0.26, 95% CI = [−0.43, −0.09], SE = 0.09, $p = 0.003$) and this trended in the same direction for women (β = −0.36, 95% CI = [−0.73, 0.01], SE = 0.19, $p = 0.057$). There was no statistically significant evidence for the association between receiving good balls and age (β = −0.13, 95% CI = [−0.31, 0.04], SE = 0.09, $p = 0.134$). In $N = 2369$ participants who completed an abbreviated compulsivity scale (20 items versus 209, see Methods[38]), we found no statistically significant evidence for an association between number of good balls received and CIT (β = 0.16, 95% CI = [−0.22, 0.53], SE = 0.19, $p = 0.416$) or AD (β = −0.06, 95% CI = [−0.43, −0.09], SE = 0.20, $p = 0.762$). In terms of hitting, older adults (β = −3.39, 95% CI = [−3.72, −3.07], SE = 0.17, $p < 0.001$), women (β = −9.48, 95% CI = [−10.16, −8.79], SE = 0.35, $p < 0.001$), and those less educated (β = −0.47, 95% CI = [−0.79, −0.16], SE = 0.16, $p = 0.003$) were less likely to hit the diamond (considering only trials with good balls). This was also the case for those with greater self-report CIT (β = −1.29, 95% CI = [−1.88, −0.71], SE = 0.30, $p < 0.001$), while AD was associated with being more likely to hit the diamond (β = 0.89, 95% CI = [0.32, 1.47], SE = 0.29, $p = 0.002$), controlling for age, gender, and education. Both model-based (β = 6.57, 95% CI = [5.58, 7.55], SE = 0.50, $p < 0.001$) and model-free (β = 4.38, 95% CI = [3.72, 5.03], SE = 0.33, $p < 0.001$) behaviours were positively associated with hitting the diamond, suggestive of a general attention/engagement effect.

The diamond hitting task was fairly challenging in that on trials where participants received good balls (and therefore a hit was possible), they hit the target on just 36.5% of shots. This would also mean that on ~27% of all trials, users received not only a good ball, but also received the additional reward of hitting a diamond, which increased their score. This additional reward could plausibly impact model-based/free behaviour by amplifying the reward signal on those trials. Another possibility however is that because the diamond moves location after it is successfully hit, this could in fact interrupt learning and so have the opposite effect. We tested this by entering Diamond Hit into the model with reward and transition on stay behaviour. We found no statistical significant evidence for a main effect of lagged diamond hit on stay behaviour (β = −0.05, 95% CI = [−0.10, 0.00], SE = 0.03, $p = 0.053$). Similarly, on trials following a diamond hit, users had reduced model-free behaviour (Reward*Diamond Hit interaction, β = −0.14, 95% CI = [−0.19, −0.09], SE = 0.03, $p < 0.001$), and reduced model-based behaviour (Reward *Transition* Diamond Hit interaction, β = −0.09, 95% CI = [−0.14, −0.04], SE = 0.03, $p < 0.001$) (Supplementary Table 10). Together these findings suggest that the receipt of a diamond functioned to impair learning, not potentiate it.

**Table 2 The impact of different task modifications on mean-level model-based estimates and its reliability.**

| | N | MBI Score | | Split-half [95%CI][a] | t/F |
|---|---|---|---|---|---|
| | | **M(SD)** | **Range** | | |
| Overall Task | | | | | n/a |
| | 5005 | 0.26 (0.33) | [−0.49−2.26] | 0.65 [0.63−0.67] | |
| Transition Ratio (Between-subject)[b] | | | | | 8.40*** |
| 80:20 | 2138 | 0.30 (0.34) | [−0.47−2.03] | 0.67 [0.64−0.69] | |
| 70:30 | 2138 | 0.22 (0.31) | [−0.45−2.16] | 0.63 [0.61−0.66] | |
| Difficulty/Order (Within-subject) | | | | | 20.16*** |
| Easy/1st Block | 5005 | 0.30 (0.32) | [−0.49−1.90] | 0.71 [0.71−0.76] | |
| Medium/2nd Block | 5005 | 0.22 (0.29) | [−0.53−1.92] | 0.68 [0.66−0.70] | |
| Reward Drift Set (Between-subject) | | | | | −5.33*** |
| Drift A/1st Block | 2395 | 0.28 (0.32) | [−0.46−1.84] | 0.74 [0.71−0.76] | |
| Drift B/1st Block | 2610 | 0.33 (0.33) | [−0.51−1.93] | 0.69 [0.66−0.71] | |

[a]Data was split using odds-evens method, reporting Pearson R.
[b]Subsample of total dataset to achieve two age, gender and education-matched samples.
***p < 0.001.
MBI Model-based Index, M Mean, SD Standard Deviation, CI Confidence Interval.

## Effect of task modifications on the external validity and reliability of model-based planning

*Transition structure.* The initial set of participants completed a version of the task with an 80:20 transition ratio (i.e., probability of purple/pink balls from the respective containers). To test if this adjustment to transition ratio impacted estimates of model-based planning, we adjusted this in-app to 70:30 (as per the classic two-step task) and continued to gather data. We subsampled our total dataset to achieve two age, gender, and education-matched samples of N = 2138 that completed the 80:20 and 70:30 versions, respectively. Participants in both groups had a main effect of Reward, Transition, and a Reward x Transition interaction (Supplementary Table 7). The 80:20 transition ratio yielded significantly greater MBI (M = 0.30, SD = 0.34) compared to those who experienced the 70:30 version (M = 0.22, SD = 0.31), t(4233.1) = 8.40, p < 0.001 (Table 2, Fig. 4a(i)). However, transition ratio structure did not affect its reliability with both versions showing similar split-half reliability (r(2136) = 0.67, p < 0.001, 95% CI = 0.64–0.69 for 80:20 and r(2136) = 0.63, p < 0.001, 95% CI = 0.61–0.66 for 70:30, Table 2).

From this sample, N = 2071 participants (80:20: N = 1020, 70:30: N = 1051) had compulsivity scores and demographic information available. A significant negative association between MBI and CIT was observed in the 70:30 group (β = −0.05, 95% CI = [−0.07, −0.03], SE = 0.01, p < 0.001) but there was no statistically significant evidence for the same relationship in the 80:20 group (β = −0.01, 95% CI = [−0.03, 0.01], SE = 0.01, p = 0.374) (Fig. 4b(i); Supplementary Table 11). This difference was confirmed by entering Transition Ratio (80:20, 70:30) into the model, where we found a significant CIT x Transition Ratio interaction (β = −0.02, 95% CI = [−0.03, −0.00], SE = 0.01, p = 0.022, Supplementary Table 12). In terms of the other individual differences, we found a significant Gender x Transition interaction in the full comparison model (β = 0.02, 95% CI = [0.01, 0.04], SE = 0.01, p = 0.009, Supplementary Table 12). This was driven by a negative association between MBI and Gender in 80:20 ratio (β = −0.04, 95% CI = [−0.06, −0.02], SE = 0.01, p = 0.001, Supplementary Table 11) but this was not statistically significant in the 70:30 group (β = 0.00, 95% CI = [−0.02, 0.03], SE = 0.01, p = 0.689, Supplementary Table 11). We found no statistical significant evidence that transition ratio effects the relationship between MBI and age (β = −0.00, 95% CI = [−0.02, 0.01], SE = 0.01, p = 0.658) or education (β = 0.00, 95% CI = [−0.01, 0.02], SE = 0.01, p = 0.874) (Supplementary Table 12).

*Difficulty and Order Effects.* Difficulty in Cannon Blast referred to how challenging the diamond shooting task was on a given trial, with harder levels requiring more timing and spatial reasoning to hit the target than easy trials (where the diamond was unobstructed or static). We hypothesised that this would put a strain on the cognitive resources required for maintaining an accurate model of the task environment and therefore impede model-based planning. As part of their first play of Cannon Blast during the Risk Factors section, all participants completed two blocks of trials; a block of 100 Easy difficulty trials (Easy-1st Block), followed by a block of 100 Medium difficulty trials (Medium-2nd Block). In each of the two blocks (Easy-1st Block, Medium-2nd Block), we observed main effects of Reward, Transition, and their interaction (Supplementary Table 7). However, we found MBI estimates were significantly larger in the Easy-1st (M = 0.30, SD = 0.32) than Medium-2nd (M = 0.22, SD = 0.29) blocks, t(5004) = 20.16, p < 0.001 (Table 2, Fig. 4a(ii)). Because difficulty and order are confounded with one another, we cannot confirm if this reduction was driven by increased task difficulty, order, or both. To isolate the effect of difficulty from order, we turned to data from the Free Play section of the app, where a subset of participants re-engaged with Cannon Blast at a chosen difficulty level for short games of 100 trials. We compared their initial play (first play) at each difficulty level (Easy, Medium) with their next play (second play) of that difficulty. In N = 689 participants who had completed two sessions of *Cannon Blast* at an Easy difficulty, we found MBI was greater at their first play (M = 0.32, SD = 0.33) compared to their second (M = 0.21, SD = 0.23), t(688) = 8.91, p < 0.001. Importantly, we did not find this difference when we repeated this analysis in N = 556 who had two sessions of play at Medium difficulty (first play (M = 0.24, SD = 0.33); second play (M = 0.25, SD = 0.33), t(555) = −0.59, p = 0.552). This suggests that our difficulty manipulation did not affect model-based planning, but that there was an effect of order – MBI estimates are greatest during the first play of Cannon Blast, which reduces and stabilises across subsequent sessions. To confirm this, we compared model-based estimates in N = 335 who had sessions at both difficulties (Easy, Medium) and at two-time points (First play, Second play). We found MBI at Easy-First play (M = 0.33, SD = 0.35) was significantly larger than all other times, including Medium-First play (M = 0.25, SD = 0.34, p = 0.002), Easy-Second play (M = 0.22, SD = 0.23, p < 0.001), and Medium-Second play (M = 0.25, SD = 0.32, p = 0.002). More evidence for the uniqueness of the first block in which people play

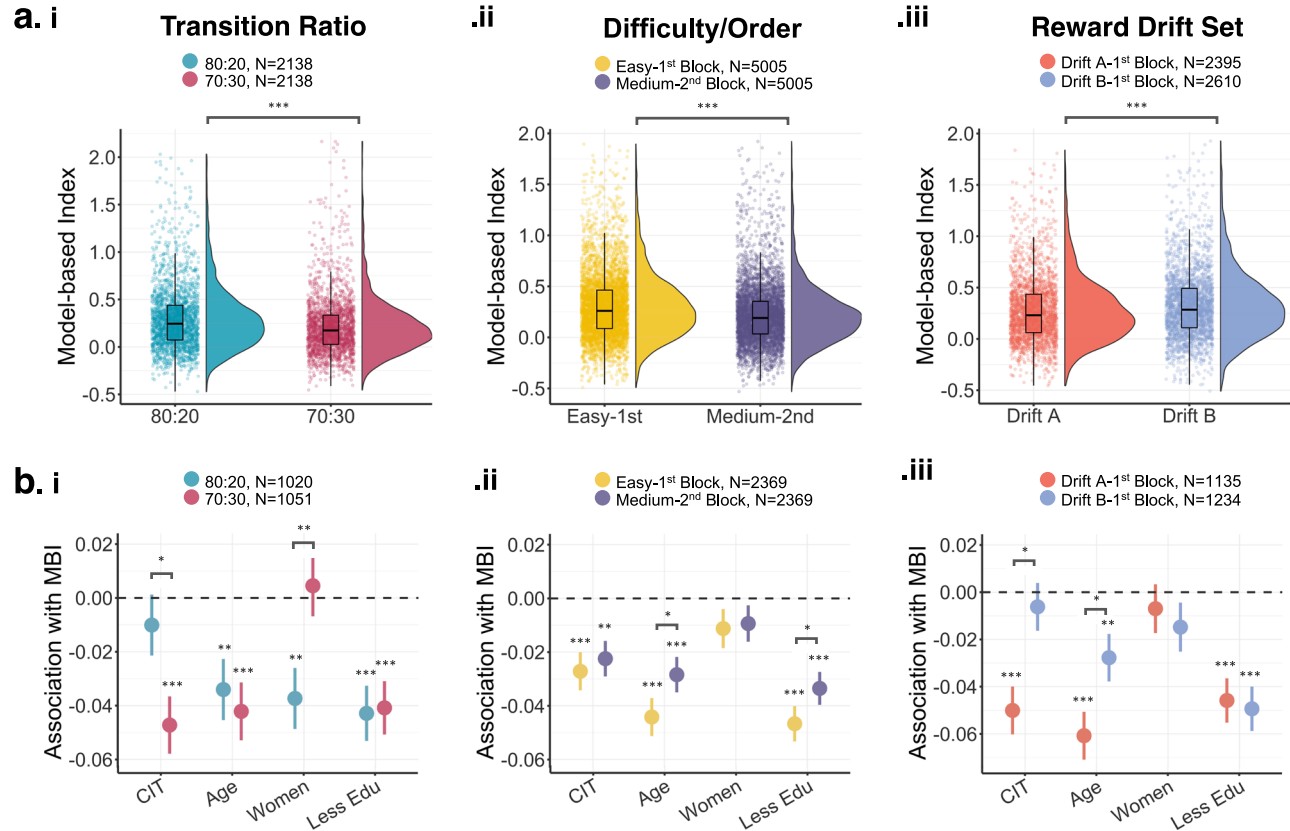

**Fig. 4 Impact of task modifications on model-based estimates and its association with compulsivity and individual differences. a** Distribution and mean value of model-based scores across (i) transition ratio, (ii) difficulty/order, and (iii) reward drift set. Mean model-based scores were larger when estimated from an 80:20 transition ratio structure compared to 70:30, in easy/1st play compared to Medium/2nd play and in those who experienced Drift set B compared to Drift set A. **b** Model-based associations with compulsivity (CIT) and individual differences (age, gender and education ('Less Edu')) across (i) transition ratio, (ii) difficulty/order, and (iii) reward drift set. The association between greater self-report compulsivity and MBI was greater when estimated from 70:30 transition ratio structure and using drift set A. Error bars reflect standard errors of the mean. *$p < 0.05$; **$p < 0.01$; $p < 0.001$***.

comes from an analysis of test re-test reliability of task sections. MBI estimated from the Easy-1st Block of the game was weakly associated with their second sessions at Easy ($N = 689$, $ICC_1 = 0.35$). The same analysis of Medium plays yield higher reliability ($N = 556$, $ICC_1 = 0.51$) (Supplementary Table 13). In terms of reliability within each block, we found comparable split-half reliability estimates (Easy-1st Block: $r(5003) = 0.72$, $p < 0.001$, 95%CI = 0.70–0.74; Medium-2nd Block $r(5003) = 0.68$, $p < 0.001$, 95%CI = 0.66–0.70, Table 2).

Having established mean-level differences across the blocks of the task, we next tested if the observed relationships with compulsivity, age, gender, and education varied across these blocks (Easy-1st Block, Medium-2nd Block). In each block, we found deficits in MBI were associated with greater compulsivity, older adults, and those with less education (Fig. 4b(ii); Supplementary Table 11). In the full comparison model where Difficulty/Order (Easy-1st Block, Medium-2nd Block) was entered into the model, we found no statistical significant evidence that Difficulty/Order impacts the association between MBI and compulsivity ($\beta = 0.00$, 95% CI = [−0.01, 0.01], SE = 0.00, $p = 0.476$). There was a significant interaction between Age × Difficulty/Order ($\beta = 0.01$, 95% CI = [0.00, 0.02], SE = 0.00, $p = 0.017$) and between Education × Difficulty/Order ($\beta = 0.01$, 95% CI = [0.00, 0.01], SE = 0.00, $p = 0.032$), such that the association between MBI and age and MBI and education were greater when estimated from the Easy-1st Block (Supplementary Table 12).

*Reward Drifts.* At each block, participants were randomly assigned to one of two drift sets (Drift A, Drift B). It is important to note that these drifts differed in more than one dimension (Fig. 1d); but descriptively, Drift A included two relatively stable reward probabilities (SD: purple = 0.053, Pink = 0.049) and relatively high reward rates (purple = 0.845, pink = 0.748), with purple balls outperforming pink, though at times there was very little evidence to distinguish the most rewarding ball colour. Drift B, in contrast, had lower overall reward rates (purple = 0.774, pink = 0.501). Purple and pink balls started out with a similar level of reinforcement and throughout the 100 trials the value of the purple steadily increased, while pink remained low (close to 0.5). Participants were randomly assigned their drifts independently across blocks, creating four possible reward drift combinations: A-A, A-B, B-A and B-B for their first 200 trials of the game. To avoid washing out effects in the mixed conditions (A-B, B-A), we compared A and B in the 1st Block only (100 trials). $N = 2139$ were randomly assigned Drift A and $N = 2345$ Drift B. In both drift sets, we observed main effects of Reward, Transition, and their interaction (Supplementary Table 7), but MBI estimates were significantly larger in those who received Drift B (M = 0.33, SD = 0.33) compared to Drift A (M = 0.23, SD = 0.31), $t(4475) = −5.33$, $p < 0.001$ (Table 2, Fig. 4a(iii)).

$N = 1135$ in the Drift A group and $N = 1234$ in Drift B group had compulsivity scores and demographic information. We found a negative association between MBI and compulsivity when estimated in those who experienced Drift A ($\beta = -0.05$, 95%

CI = [−0.07, −0.03], SE = 0.01, p < 0.001, Supplementary Table 11), but no statistically significant evidence for this association when estimated from Drift B (β = −0.01, 95% CI = [−0.03, 0.01], SE = 0.01, p = 0.540). In a full comparison model where Drift (Drift A, Drift B) was entered into the model, this difference was confirmed by a significant CIT x Drift interaction (β = 0.02, 95% CI = [0.01, 0.04], SE = 0.01, p = 0.002, Supplementary Table 12). We also observed a significant Age x Drift set interaction (β = 0.02, 95% CI = [0.00, 0.03], SE = 0.01, p = 0.024, Supplementary Table 12), with estimates between age and MBI greater when estimated from those in Drift A.

*Trial Number.* To test the impact of increasing the total number of trials used to estimate MBI per participant, we generated each participant's MBI several times, starting with a participant's first 25 trials and increasing by 25 trials in each iteration, until 300

trials per participant was reached. In all cumulative trial number sets, we observed main effects of Reward, Transition, and their interaction (Supplementary Table 15). This was done in the sub-sample of participants who had 300 trials of Cannon Blast (N = 716). We found there was overall a reduction in MBI as trial number increased (β = −0.02, 95% CI = [−0.04, 0.00], SE = 0.01, p = 0.004, Figure 5ai, Supplementary Table 16). As expected, increasing the amount of data collected per participant improved internal consistency of MBI estimates; reliability at 25 trials was r(714) = 0.41, p < 0.001, 95% CI = 0.35−0.47 and increased to r(714) = 0.71, p < 0.001, 95% CI = 0.68−0.75 at 300 trials (Figure 5bi; Supplementary Table 14). However, in terms of external validity, there was no statistical evidence that the association between MBI and compulsivity was impacted by additional trials (CIT x Trial Number: β = −0.00, 95% CI = [−0.00, 0.00], SE = 0.00, p = 0.819, Figure 5ci, Supplementary Table 17), which was

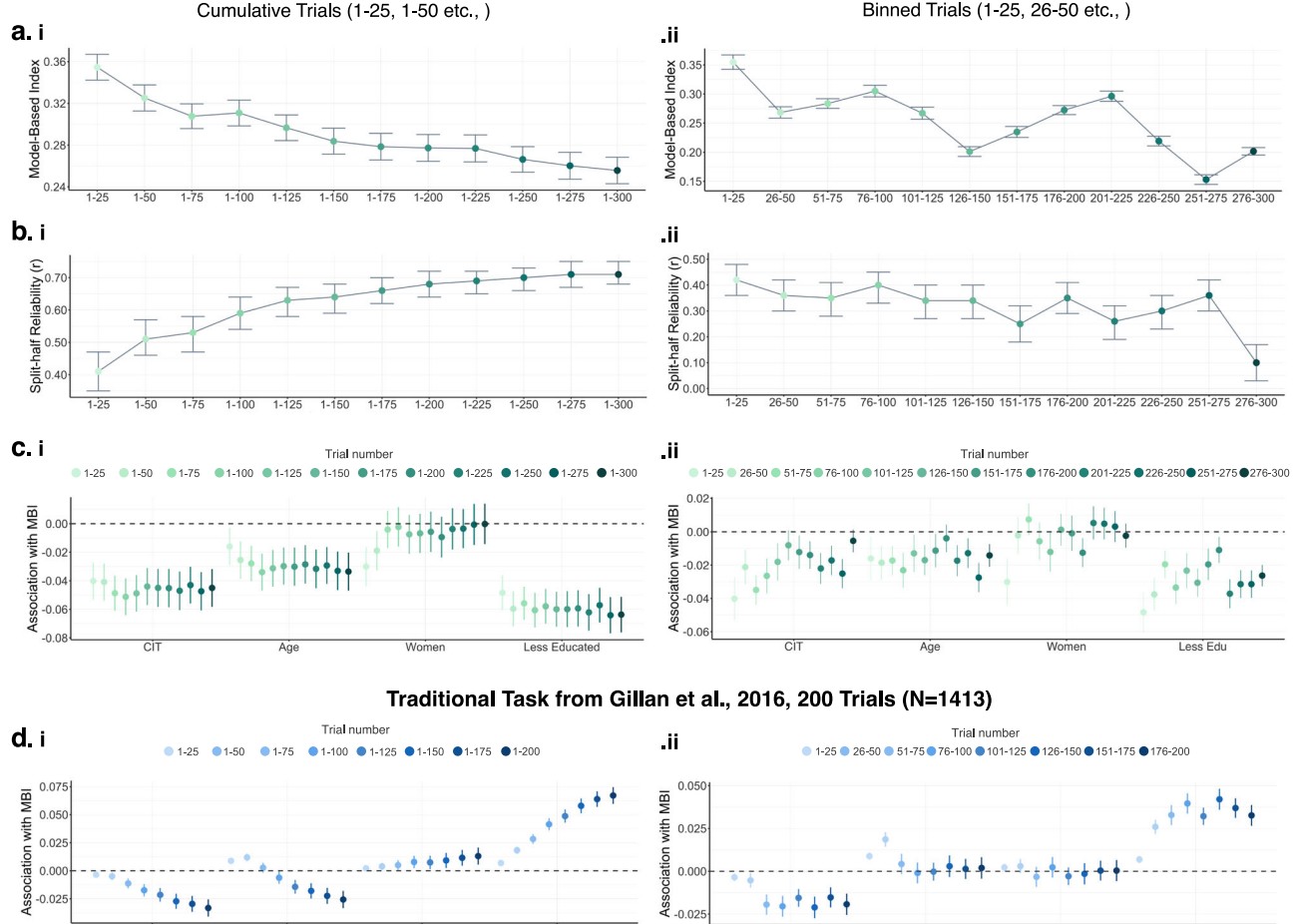

**Fig. 5 The impact of increasing number of trials on mean-level model-based estimates, its reliability and their association with individual differences (N = 716). a** Mean MBI estimated with (i) cumulative trials i.e., increasing trial numbers sequentially by 25 at a time, until 300 trials per participant was reached, and (ii) binned trials i.e., bins of 25 trials sampled sequentially through the task, in chronological order. Mean-level MBI decreased with more trials. Error bars reflect standard errors of the mean. **b** Split-half reliability co-efficient as a function of increasing trial number using (i) cumulative trials and (ii) binned trials. Using the cumulative trials, reliability of the MB estimate increased with additional trial collected per participant. Error bars reflect 95% confidence intervals **c** Model-based associations with compulsivity, age, gender and education using (i) cumulative trials and (ii) binned trials. Increasing the number of trials per participant did not significantly increase the association between model-based planning and individual differences. No statistical significant difference was found between model-based associations with compulsivity when estimated at participants' first 25 vs 300 trials. Model-based associations with age and education became stronger with the addition of trials while the association between model-based and gender reduced. Data from the binned analysis demonstrated that this effect is in part driven by stronger signal in earlier versus later trials. **d** We repeated this analysis using a publicly available dataset of N = 1413 individuals who completed the traditional two-step task (200 trials). Here we found (i) the association between model-based planning and individual differences increased as trials collected per participant increased and (ii) this effect is driven by later trials compared to earlier trials. For **c** and **d**, error bars reflect standard errors of the mean.

significant when estimated with as few as 25 trials ($\beta = -0.04$, 95% CI = $[-0.07, -0.02]$, SE = 0.01, $p = 0.002$, Supplementary Table 16). There were significant interactions between Age x Trial Number ($\beta = -0.01$, 95% CI = $[-0.00, -0.00]$, SE = 0.00, $p = 0.044$), Gender x Trial Number ($\beta = 0.00$, 95% CI = $[0.00, 0.00]$, SE = 0.01, $p. < 0.001$) and Education x Trial Number ($\beta = 0.00$, 95% CI = $[-0.00, 0.00]$, SE = 0.01, $p = 0.042$) (Supplementary Table 17). The association between both age and education and MBI became stronger with the addition of more trials while the association between gender and MBI reduced.

Next, we wanted to test if any specific collection of trials were driving these effects. To do this, we repeated the above analyses using bins of 25 trials (0–25, 26–50, etc). Similar to cumulative trials, we found an overall reduction in MBI for later trial bins (Figure 5aii). Reliability estimates were relatively stable across the bins of trials (Figure 5bii). But associations between MB and CIT reduce as the task progresses ($\beta = 0.002$, 95% CI = $[0.00, 0.00]$, SE = 0.00, $p = 0.010$, Figure 5cii). To test if this pattern is specific to our task, we carried out the same analysis on data from the traditional task in a previously published study ($N = 1413$ who completed the traditional two-step task remotely online)[3]. In the traditional task, participants must learn the transition probabilities through trial and error (unlike our task where they are shown on-screen throughout), so it follows that the early trials should not be informative for assessing model-based planning. Indeed this was the case; in the traditional task, associations with CIT increase with more trials when examined cumulatively (Figure 5di) and associations with CIT were absent from the first 2 bins of trials, and only become significant later (Figure 5dii).

## Discussion

We developed a smartphone-based diamond-shooting game capable of assessing model-based planning in out-of-the-lab unconstrained settings. We demonstrated that estimates derived from the game are valid by replicating previously established correlates of model-based planning: age[24], gender[3], education (similar to previous work on IQ[3] and processing speed[25]) and compulsivity[3–5,7,15] in a large sample of citizen scientists and by demonstrating comparable psychometrics to the traditional. We then used this task to tackle a question posed by recent studies[11–15,41]: are associations between compulsivity and model-based planning dependent on key aspects of the task's design? In a series of within and between-participant experiments, we found that some task parameters (i.e., transition and reward drifts) are associated with better capture of individual differences. The association between model-based planning and compulsivity was greater when using a less deterministic transition ratio structure (70:30 compared to 80:20) and that this association also depends on the specific nature of the drifting reward probabilities used. We additionally showed that previously established effects could be observed in as few as 25 trials, and that increasing trial numbers had little impact. We did not find the association between model-based planning and compulsivity significantly differed between manipulations made to task difficulty. However, this could be a limitation of our task design, which used a relatively weak dual-task manipulation. Specifically, difficulty in this context referred to the extra demands placed on participants to shoot the diamond on harder levels, which required taking into account the geometry of the screen and timing the shot. Perhaps crucially, this demand occurs after the model-based update had occurred and participants have presumably already decided which container to fire from. A more potent dual-task demand would impact the update itself[10].

Previous findings suggest that model-based planning problems in compulsivity arise from issues in building accurate mental models of action-outcome contingencies[7], which may arise from problems with learning probabilistic action-state transitions[8], and assigning credit to actions[42]. Recent work extended this idea, finding that differences between OCD patients and controls are largest when instructions are absent and individual must learn the structure of the task from scratch[14]. In our task, the relationship between an action (choosing left or right) and the resulting state (ball colour) was visible on-screen throughout the game as the proportion of balls displayed in each container. Despite this, we found the characteristic reduction in model-based planning in compulsive individuals. This suggests that the mechanisms underlying this relationship reflect more than a failure to learn about the statistical properties of action-state transitions through experience. One possibility is that in compulsivity, broader issues in executive function might cause individuals to simplify tasks to avoid overload of working memory or other finite cognitive resources[43]. This notion is supported by the finding from the present study that the association between model-based planning and compulsivity is greater when the action-state transitions are more uncertain, i.e., less deterministic (i.e., 70:30), and model-based planning is more effortful to employ (i.e., requires more win-switch and lose-stay actions).

In our task, the drift that had the stronger association between model-based planning and compulsivity had higher overall reward rates, a fairly stable time course and relatively low distinguishability between the two states compared to a drift rate with lower over-reward rates and higher distinguishability between states. On a similar note to the above point, it could be that the cost-benefit of engaging in model-based behaviours is not worthwhile in environments where there isn't a noticeable difference in reward magnitudes of outcomes. However, because the drifts used differed on more than one dimension, future research is needed to understand and further optimise the selection of drifts. This finding is nonetheless important, as studies often vary in the drifts employed with no established best-practice. One suggestion is to revisit these analyses using all ten possible drift sets available in the app, once we have collected a sufficient amount of data per drift set. Any candidate drift properties (e.g., boundaries, variance, drift rate) that affect the relationship between model-based planning and compulsivity can later be more systematically confirmed in new drift sequences optimised to maximally compare relevant dimensions.

In many areas of psychology, collecting more data per participant can improve the reliability and therefore the strength of associations with individual difference measures[44]. However, it is important to note that the relationship between trial number and reliability is non-linear; reliability improves steeply until it reaches asymptote[21,45]. In line with this, we observed the split-half reliability of model-based estimates increased with the addition of trials however began to taper off at around 125 trials. A second issue when considering trial number is that additional trials can change the nature of the measurement, not just its reliability. Depending on the design, behaviour on tasks that involve some element of learning can signal different processes at different stages of the task. That is, the initial trials where a participant explores and becomes familiar with the task can reflect a different cognitive process than later trials when a participant is updating and executing what they have learned. This is consistent with what we observed in the present study; model-based planning was greatest during participants first block of trials and decreased steadily over time. and test-retest estimates were lower for the 1st play than the 2nd. The learning that takes place in the first trials of the task may mean that those trials carry maximal variation in the process of interest, and so counterintuitively it could mean that additional trials in fact reduce the magnitude of group differences. This finding is important for

future research studies considering to use tasks with elements of reinforcement learning measured over multiple time points for example pre- and post-treatment. To make these sessions more homogenous, longer intervals between sessions may be required to restore the initial block effect. In the present study, we found that additional trials did not affect the strength of the observed relationship with compulsivity, which was apparent with just 25 trials of data. More trials did however modestly improve the effect sizes for the link between MBI and age and education, but this was mostly driven by gains from 25 to 100 trials. This early signal in the task stands in contrast to the traditional version, where we show that early trials do not carry much individual difference signal. We speculate that this difference is because in the traditional task, participants must learn the transition probabilities through trial and error (and so model-based planning cannot be executed from trial 1), whereas in Cannon Blast, the probabilities are displayed on-screen. This simple change in design may have major implications for future studies aiming to reduce the trials required to estimate model-based planning.

We found that model-free estimates were associated with the same individual and clinical associations as model-based planning. This is in line with a prior study that also observed reduced model-free learning in compulsivity[5], but it is important to note that the majority of studies have not observed this[3,4,6]. In fact, conceptually, one might expect model-free behaviour to be enhanced in compulsivity, given the popular framework that describes their trade-off[1]. This apparent inconsistency follows a growing literature that has raised doubts about what the model-free estimates truly represent. Prior work has shown that model-free learning is not related to the gold-standard index of habit, and performance at a devaluation test[9] and as we report here, studies consistently show a positive correlation between model-based and model-free estimates[3,7]. Recent work suggests model-free behaviour on this task may be better understood as a different form of model-based choice, that we fail to model accurately[13]. In line with this, Konovalov and Krajbich[46] found that model-free participants made more fixations prior to choice, indicating choice deliberation rather than habitual selection. In the present study, we embedded the classic two-step task within a game, that is, we made model-free and model-based behaviour operate in the service of a reward (gaining good balls), which was itself only valuable insofar as it served the higher-order goal of shooting diamonds. This meant that model-free behaviour was no longer the lowest level of engagement with the task and it is therefore possible that those with the lowest cognitive capacity may have focused exclusively on the primary task of shooting the diamonds, adding new model-based signal to the model-free estimate. Indeed, the cross-task correlations in the present study support this idea as model-free measures from Cannon blast were positively associated to both model-free and model-based estimates measured from the traditional task.

There is increasing evidence that true correlations between cognitive test performance and individual differences in mental health are small. This is increasingly recognised to be the case for various levels of analysis in psychiatry, such as neuroimaging[47], studies of environmental risk[48], and genetics[49]. Small effects can nonetheless have big impact[50], if tackled at a population level, from a public health perspective. But in order to estimate these effects and interpret them accurately, we need larger samples in our research studies. One way to achieve this is by taking our assessments out of the lab and into daily life through gamification and smartphones[51,52]. Our results add to the growing evidence that smartphones can deliver valid cognitive tests data with clinical implications[53–55]. In this paper, we emphasise another advantage of smartphone science in the cognitive space: it facilitates AB testing so we can systematically begin to improve the psychometric properties of our tasks and optimise them for specific clinical populations[18]. Another key advantage of this method is that it leverages citizen scientists, rather than exclusively relying on university students. This results in a more diverse sample and makes research more accessible to those living farther from research centres or unable to attend during working hours. Other forms of remote assessment, such as Amazon's Mechanical Turk (MTurk), connect users to studies for payment. However, recently these platforms have been at the mercy of increasing issues with data quality[56,57]. In citizen science, incentives of researcher and participant may be better aligned and this may have a positive impact on data quality[58].

**Limitations**. This research is not without limitations to consider. Taking cognitive and clinical assessments out of controlled laboratory settings and into the noisy real world can introduce concerns related to data quality. For example, task performance and self-report scores may be affected by lapses in attention, distractions or careless responding and there is a risk that this creates spurious or inflated associations between cognitive performance and mental health symptomatology[59]. To mitigate this, participants in our study are not tied to specific time constraints and have the freedom to engage with assessments suited to stop the game if interrupted, and are not financially induced to participate. A second limitation concerns sampling bias; smartphone science in some respects helps us to tackle issues with small and unrepresentative samples in psychology research, but it too comes with its own biases. Participants in this study self-selected to engage with this research and did so without remuneration. They may carry certain characteristics that are not representative of the broader population, particularly compared to those with low digital literacy or who don't have internet access.

## Conclusions
In a brief smartphone game, where participants shot at diamonds, we replicated robust associations with socio-demographics and compulsivity at-scale, and demonstrate canonical effects of model-based behaviour with psychometric properties similar to its traditional version. Overall, we present evidence that smartphone science opens the door to data-driven task-optimization, increasing their protential be translated into clinical decision tools in the future, bringing research into practice.

## Data availability
The data sourced from previously publicly available data sets can be found at https://osf.io/usdgt/[3,40] and https://osf.io/mx9kf/[7,26]. The processed data that supports the findings of this study are publicly available at https://osf.io/arhng/ without restrictions. Due to the sensitive nature of the unprocessed data generated in this study (i.e., responses to mental health questionnaire items) and to comply with data protection regulations, the unprocessed data are not shared publicly. These data can be made available from the authors upon request through the corresponding author.

## Code availability
Code to reproduce the findings and figures are available at https://osf.io/arhng/.

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

## Acknowledgements

This work was supported by a European Research Council (ERC) Starting Grant (ERC-H2020-HABIT) awarded to Claire M. Gillan. The Neureka app is funded by

the European Research Council (ERC-H2020-HABIT), Global Brain Health Institute (18GPA02), and Science Frontiers Ireland (19/FFP/6418) awarded to Claire M. Gillan. The funding bodies had no role in the study design, data collection and analysis, decision to publish or preparation of the manuscript.

## Author contributions

K.R.D. designed the study, worked on the development of the smartphone application, carried out the experiments, analysed the data, contributed to the interpretation and wrote the manuscript. V.M.B. contributed to the interpretation of the results. R.B.P. contributed to the interpretation of the results. E.G. worked on the development of the smartphone application. A.P. worked on the development of the smartphone application. A.K.H. worked on the development of the smartphone application. C.M.G. designed the study, worked on the development of the smartphone application, analysed the data, contributed to the interpretation and supervised the writing of the manuscript.

## Competing interests

Vanessa M. Brown declares the following competing interests: received consulting fees from Aya Technologies. Kelly R. Donegan, Rebecca B. Price, Eoghan Gallagher, Andrew Pringle, Anna K. Hanlon and Claire M. Gillan all declare to have no competing interest.
