## [Peer Review File · Communications Psychology]

5th May 23

Dear Claire,

Thank you for your patience during the peer-review process. I am sorry for the delay in returning to you with a decision, which resulted from reviewer (un)availability.

Your manuscript titled "Using smartphones to optimise and scale-up the assessment of model-based planning" has now been seen by 3 reviewers, and I include their comments at the end of this message.

The referees find your work of interest, but raised some important points. We are interested in the possibility of publishing your study in Communications Psychology, but would like to consider your responses to these concerns and assess a revised manuscript before we make a final decision on publication.

We therefore invite you to revise and resubmit your manuscript, along with a point-by-point response to the reviewers. Please highlight all changes in the manuscript text file.

Editorially, we consider it necessary that your revision addresses the concerns about potential alternative explanations raised by Reviewers #1 and #2, improves the clarity of reporting (esp. mentioned by Reviewers #2 and #3) and ensures that all caveats and limitations on generalizability of the results are suitably discussed.

As you address Reviewer #3's concerns regarding data protection, please ensure that you include an appropriate data availability statement (more details are included below).

Please use the following link to submit your revised manuscript, point-by-point response to the referees' comments (which should be in a separate document to any cover letter) and the completed checklist:

[link redacted]

We hope to receive your revised paper within 8-12 weeks; please let us know if you aren't able to submit it within this time so that we can discuss how best to proceed. If we don't hear from you, and the revision process takes significantly longer, we may close your file. In this event, we will still be happy to reconsider your paper at a later date, provided it still presents a significant contribution to the literature at that stage.

Please do not hesitate to contact me if you have any questions or would like to discuss these revisions further. We look forward to seeing the revised manuscript and thank you for the opportunity to review your work.

Best wishes

Marike

Marike Schiffer, PhD
Chief Editor
Communications Psychology

EDITORIAL POLICIES AND FORMATTING

Editorial Policy: [Policy requirements](https://www.nature.com/documents/nr-editorial-policy-checklist.pdf) (Download the link to your computer as a PDF.)

Furthermore, please align your manuscript with our format requirements, which are summarized on the following checklist:

[Communications Psychology formatting checklist](https://www.nature.com/documents/commsj-style-formatting-checklist-review-perspective.pdf)

and also in our style and formatting guide [Communications Psychology formatting guide](https://www.nature.com/documents/commspsychol-style-formatting-guide-accept.pdf) .

* **CODE AVAILABILITY:** All Communications Psychology manuscripts must include a section titled "Code Availability" at the end of the methods section. In the event of publication, we require that the custom analysis code supporting your conclusions is made available in a publicly accessible repository; at publication, we ask you to choose a repository that provides a DOI for the code; the link to the repository and the DOI will need to be included in the Code Availability statement. Publication as Supplementary Information will not suffice. We ask you to prepare code at this stage, to avoid delays later on in the process.

*** DATA AVAILABILITY:**

All Communications Psychology manuscripts must include a section titled "Data Availability" at the end of the Methods section or main text (if no Methods). More information on this policy, is available at <http://www.nature.com/authors/policies/data/data-availability-statements-data-citations.pdf>.

At a minimum the Data availability statement must explain how the data can be obtained and whether there are any restrictions on data sharing. Communications Psychology strongly endorses open sharing of data. If you do make your data openly available, please include in the statement:

We recommend submitting the data to discipline-specific, community-recognized repositories, where possible and a list of recommended repositories is provided at <http://www.nature.com/sdata/policies/repositories>.

If a community resource is unavailable, data can be submitted to generalist repositories such as [figshare](https://figshare.com/) or [Dryad Digital Repository](http://datadryad.org/). Please provide a unique identifier for the data (for example a DOI or a permanent URL) in the data availability statement, if possible. If the repository does not provide identifiers, we encourage authors to supply the search terms that will return the data. For data that have been obtained from publicly available sources, please provide a URL and the specific data product name in the data availability statement. Data with a DOI should be further cited in the methods reference section.

REVIEWERS' EXPERTISE:

Reviewer #1 reinforcement learning, planning
Reviewer #2 reinforcement learning, planning
Reviewer #3 decision-making, smartphone-based research

REVIEWERS' COMMENTS:

Reviewer #1 (Remarks to the Author):

I think this is a very important and useful piece of work. The authors look at the key theoretical construct of model-free vs. model-based reinforcement learning; they take the canonical (2-step) task to measure this, and developed an gamified smartphone version for it. They demonstrate with a

very large sample size that model-based and model-free decision strategies can be assessed in real-world contexts using this simple and presumably fun smartphone game. They establish both reliability and validity of the task measures, where they show they can replicate previous findings of how key task measures relate to external variables (such as e.g., compulsivity). The large sample size allows them to test which task features/configurations provide highest reliability and validity, e.g., suggesting that a very small number of early trials already has high external validity for some measures. One very important contribution that I see with this work is, that it will allow to study key model-free/-based learning systems at high time resolution (i.e., repeated assessments e.g., over weeks) with large sample sizes in real-world settings, such as in (sub)clinical populations. As such, I think that the work provides an important contribution to the literature, and I anticipate that it will have considerable impact.

I have the following comments:

Most importantly, as far as I can see, the authors used the chosen ball to define whether reward was obtained (i.e., a „good ball“, which did not explode early) or not (i.e., a „bad ball“, which exploded too early and therefore couldn't hit the diamond). However, as the authors write, drawing a „good“ ball did not guarantee that the diamond was actually hit and a win occurred. Thus, a possible alternative definition of reward might consider whether the diamond was actually hit and whether subjects actually received reward. I am wondering whether their relatively abstract definition of „reward“ might explain some of the counter-intuitive findings relating to the model-free index, since it seems conceivable that the model-free system might (at least also) care about actual rewards (diamond hit) rather than only about „good“ balls that don't actually lead to reward/points won.

Concerning the drift rates: it seems that the authors decided a priori for a certain selection of „random walks“, and they investigate task measures for different selected versions of these „random walks“. I was wondering how these precise „random walks“ were derived? As the authors state - they seem to differ on various dimensions; thus, I don't see how they allow drawing more general conclusions. An alternative approach that the authors might consider discussing as potential future work might be to systematically vary certain aspects of the random walks (e.g., average step size, reflecting boundary range, average reward probability, or longer-term drifts; ideally computationally derived) and test their impact on task reliability/validity. However, I think this is only a minor aspect of the work, and doesn't impact on its generally high importance.

Daniel Schad

Reviewer #2 (Remarks to the Author):

Donegan and colleagues report their findings of two experiments that employ a novel, gamified, version of the two-step task, which measures the degree to which people engage in model-based vs model-free reinforcement learning. They show that the task has reasonable cross task reliability with a more conventional version of the task. Moreover, they show that the new task task has a few advantages over the original, with the most noteworthy insight being that reliable estimates can be obtained with relatively few trials. In addition, they claim to replicate a host of findings, including that concurrent task demands and changes to task structure can decrease model-based control, and that several individual differences (age, sex, IQ) predict use of model-based control.

I think this is an interesting paper, with a cool, inventive, and novel way to study model-based decision making. There is a large potential for this version of the task to transform the way people study model-based reinforcement learning. However, at the same time, I have several concerns and questions that would need to be addressed before this paper is ready for publication.

First, I am not particularly convinced about the difficulty/task-order analyses. If I understand correctly, the authors are trying to use these to replicate Otto et al. (2013), who showed that maintaining information in working memory during decision making reduced model-based control. It's not clear to me that changes in difficulty here constitute the same manipulation. For example, if I understand the supplied videos, all of these difficulty manipulations are about making the target harder to hit. Therefore, the participant can first decide which "box" they will choose (based on recent information), and then start aiming with the cannon. Note that in this scenario, all model-based planning will have already occurred before aiming difficulty can influence the decision. A more direct replication would have introduced demands that irrefutably would be present throughout the trial.

Second, and related to this, the task introduces a novel skill-related component, and the authors largely gloss over the implications of this in their task. What's striking to me about this is that participants only hit targets on 36.5% of all shots on which they received a good ball. This greatly reduces the efficacy of any of the RL systems — their outputs typically do not lead to a win. In fact, it looks like participant only score on ~27% of trials ($75\% \times 36.5\%$). Isn't it possible that this induces the reduction in model-based control over time? That is, with such a low hit rate, and with computational demands for the MB system, participants learn that it's not worth it to plan the next choice of ball: it will like lead to a loss anyway. Another unexplored facet of this additional feature in the task is that there are now essentially two forms of feedback: (1) whether or not a "good" ball was drawn, and (2) whether or not a diamond was scored. Does this second reward signal contribute to subsequent choice? It's easy to see how a model-free system would update actions values according to this second reward signal. Apart from the section on "broader patterns" (which mainly reports correlations), I did not see an in-depth analysis on this feature of the task.

My third worry revolves around two recent studies by Feher - Da Silva and Hare in NHB (2020, 2023), that seem to demonstrate that signatures of model-free control are actually better understood as people applying model-based control over faulty models. To be honest, I am skeptical of those results, and I think that the current results could deliver evidence against this interpretation. That being said, I think the authors could do a more thorough job in describing how their design mitigates these concerns.

Fourth, it appears that for all drifting reward probabilities, the purple option is almost always better. This strikes me as odd, isn't the idea of the drifting probabilities that it should keep people on their toes about which option is better? Right now, it seems like people could have just clued in too that the purple is the superior one, and to let this guide choice (potentially reducing the MB influence). Why did the authors not simply implement truly randomly drifting rewards. The authors claim that the typical two-step task only includes one drift sequence, but many modern papers let rewards randomly over time for each participant (one of many papers that do this: Bolenz et al., 2022, Scientific Reports).

I found the section on difficulty and task-order hard to read. I think this is partly because these two

components are confounded, but also because it never becomes exactly clear what the authors conclude from the findings. This is partly because it's unclear what the "free play" session entails, and how these involve "first" and "second" blocks. Moreover, it is unclear what conclusions the authors draw from the analyses of data in this phase? Do they think task difficulty decreases model-based control? Do they think task order influences model-based control? Or are these components so confounded that it's truly hard to determine what can be concluded. I think this section would benefit from a more explicit description of the conclusions.

If I understand correctly, "bad balls" are observed for a shorter period of time than "good balls". This appears to change demands for wins and losses (e.g., outcomes need to be processed faster than losses). Have the authors considered the effect of this on their task? For example, during computational modeling, they could include separate learning rates for wins and losses. This is already an established practice for RL tasks, but here it seems particularly important to account for this change in the paradigm.

For the trial number analysis, it is not clear whether the graph shows that simply increasing the number of trials entered in to the model-fitting procedure reduces the estimate the model-based component, or whether MB control shows a reduction over time. That is, for the data points labeled "25 trials", are these the first twenty five trials in the task, or just 25 randomly picked trials? Either of these options have considerable caveats. If the former, then the reduction in MB control over trials is not surprising, because later trials involved more difficult task aspects. If the latter, then the conclusions drawn from this analysis, that you can get robust estimates with few trials, is a bit misleading because this scenario is not the same as what would happen if one only runs 25-trial two-step tasks (where the trials are consecutive). This section requires clarification, and the authors should hedge the conclusions they draw from this section.

Minor concerns:

I am genuinely confused why the participants in the second phase are called "citizen scientists". If I understand correctly, these people just took part in the task, but did not conduct any of the research? Maybe this is a term that the particular app developed to entice people to take part in their studies, but I don't think it fits a scientific publication.

The authors mention "model-free planning" (page 8), and that seems like a contradiction in terms.

Reviewer #3 (Remarks to the Author):

I was asked to comment on data protection issues and ethical issues regarding the smartphone-based approach used in this work – but could not find sufficient information in the paper. Given the importance of these issues, particularly regarding the fact that sensitive (mental-health-related) data are collected, I consider it important that the authors describe data protection procedures, terms of data use, and what participants gave their consent for in detail. Also, information of whether there was an ethics committee approval of the current study is essential. I found some statement that "Neureka has received ethical approval from the School of Psychology Committee on Research Ethics and is fully GDPR compliant." on the internet, but it should be made clear in the paper in

which way the present work is covered by this general approval.

Otherwise, I think that this an important and methodologically sound study. Methods are well described, results clearly presented, and conclusions well supported.

Particularly the results on the impact of increasing the number of trials on parameter estimates, their reliability, and associations with other variables are highly interesting and generally relevant for researchers who want to create online versions of certain tasks and (could) have them done extensively by participants.

I just have a few minor comments, primarily on further information that could be provided.

It may be interesting to get some more information on how the difficulty of the shooting task was set. Was this somehow calibrated so to be maximally motivating for participants? The authors write that “the task was fairly challenging”, but it is not clear whether this was intended and whether it may be too difficult for certain groups (particularly older adults).

N = 57 in Exp 1 (targeted sample size was a minimum of 50) may be sufficient to get medium-sized effects significant, but is not a large sample to estimate the correlation of the different task versions with high precision. What is the CI of the reported correlation of .40?

I would recommend using one alpha level throughout (i.e., refrain from using different “significance levels”).

Regarding the matched samples (reported in Table S7), it would be interesting to get some descriptive information (means and SDs of the 3 matching variables in the two samples) about how well the achieved matching was.

REFEREES' REVISIONS & AUTHORS' RESPONSES FOR COMMSPSYCHOL-23-0053-T

REVIEWER #1

"I think this is a very important and useful piece of work. The authors look at the key theoretical construct of model-free vs. model-based reinforcement learning; they take the canonical (2-step) task to measure this and developed a gamified smartphone version for it. They demonstrate with a very large sample size that model-based and model-free decision strategies can be assessed in real-world contexts using this simple and presumably fun smartphone game. They establish both reliability and validity of the task measures, where they show they can replicate previous findings of how key task measures relate to external variables (such as e.g., compulsivity). The large sample size allows them to test which task features/configurations provide highest reliability and validity, e.g., suggesting that a very small number of early trials already has high external validity for some measures. One very important contribution that I see with this work is, that it will allow to study key model-free/-based learning systems at high time resolution (i.e., repeated assessments e.g., over weeks) with large sample sizes in real-world settings, such as in (sub)clinical populations. As such, I think that the work provides an important contribution to the literature, and I anticipate that it will have considerable impact.

I have the following comments"

Thank you for your review of our manuscript. We appreciate your positive feedback. Your suggestions for improvements are carefully considered below.

"Most importantly, as far as I can see, the authors used the chosen ball to define whether reward was obtained (i.e., a "good ball", which did not explode early) or not (i.e., a "bad ball", which exploded too early and therefore couldn't hit the diamond). However, as the authors write, drawing a "good" ball did not guarantee that the diamond was actually hit, and a win occurred. Thus, a possible alternative definition of reward might consider whether the diamond was actually hit and whether subjects actually received reward. I am wondering whether their relatively abstract definition of "reward" might explain some of the counter-intuitive findings relating to the model-free index, since it seems conceivable that the model-free system might (at least also) care about actual rewards (diamond hit) rather than only about "good" balls that don't actually lead to reward/points won."

Thank you for highlighting the distinction between the two potential forms of reward in Cannon Blast. We have made updates in the text (Page 5, Line 182-184) to clarify this difference between a reward of a good ball to shoot with and the reward of hitting the diamond for a point increase.

"This means there are two potential forms of reward in this task – getting a 'good ball' and hitting the diamond. For clarity, we define 'Reward' in Cannon Blast as the former, but unpack the impact of the latter on choice in our later analyses."

Furthermore, we completed additional analyses to address the point raised. We tested the impact of this secondary reward signal on stay behaviour by adding it as an additional variable into the model along with reward and transition. Contrary to the reviewer's expectation, we found that the main effect of reward remained when this additional 'diamond hit' event was entered into the model ($\beta = .43$, $se = .03$, $p < .001$), and crucially, model-based, model-free and stay behaviour all decreased after a diamond hit. We believe that this is in part an artefact of the task design, whereby a successful diamond hit means that the diamond moves location on the next trial and this likely interrupts learning a bit. We have added these results into the manuscript (Page 8, Line 286-298 & Supplementary Page 16).

"This would also mean that on ~27% of all trials, users received not only a 'good ball', but also hit a diamond, which increased their score. This additional reward could impact model-based/free behaviour by amplifying the reward signal on those trials. Another possibility however is that because the diamond moves location after it is successfully hit,

this could in fact interrupt learning and so have the opposite effect. We tested this by entering 'Diamond Hit' into the model with reward and transition on stay behaviour. We found a negative main effect of lagged diamond hit on stay behaviour, though this failed to reach statistical significance ($\beta = -.05$ (.03), $p = .053$) (Table S10). Similarly, on trials following a diamond hit, users had reduced model-free behaviour (Reward*Diamond Hit interaction, $\beta = -.14$ (.03), $p < .011$), and reduced model-based behaviour (Reward*Transition*Diamond Hit interaction, $\beta = -.09$ (.03), $p < .011$). Together these findings suggest that the receipt of a diamond functioned to impair learning, not potentiate it."

Table S10. Linear Regressions Predicting Stay Behaviour by Reward, Transition and Diamond Hit on the Previous Trial

	B (SE)	z	p
Intercept	1.38 (.03)	49.33	<.001
Reward	.43 (.03)	16.22	<.001
Transition	.01 (.03)	.42	.676
Diamond Hit	-.05 (.03)	-1.94	.053
Reward * Transition	.19 (.03)	7.22	<.001
Reward * Diamond Hit	-.14 (.03)	-5.39	<.001
Transition * Diamond Hit	.08 (.03)	3.10	.002
Reward * Transition * Diamond Hit	-.09 (.03)	-3.35	<.001

Variable coded as: Reward; No Reward, 1; -1, Common; Rare, 1; -1, Hit; Miss, 1; -1

"Concerning the drift rates: it seems that the authors decided a priori for a certain selection of "random walks", and they investigate task measures for different selected versions of these "random walks". I was wondering how these precise "random walks" were derived? As the authors state - they seem to differ on various dimensions; thus, I don't see how they allow drawing more general conclusions. An alternative approach that the authors might consider discussing as potential future work might be to systematically vary certain aspects of the random walks (e.g., average step size, reflecting boundary range, average reward probability, or longer-term drifts; ideally computationally derived) and test their impact on task reliability/validity. However, I think this is only a minor aspect of the work and doesn't impact on its generally high importance."

Thank you for your thoughts. Indeed, we regret that we did not vary the drifts more systematically and agree this should be a focus for future work. Several candidate drifts were derived using a python script implementing a Gaussian random walk with boundaries and we selected 10 from this that represented a 'qualitatively distinct' range of cases – subjectively selected by the research team on this basis. Of those 10, we used two for the primary gameplay and counterbalanced them. And in subsequent plays, we randomly selected one of the 10 walks for every 100 trial block they played.

Our aim in doing this was to get some preliminary data to test if different drift sequences has an important impact on mean-level model-based estimates, as previous studies have suggested (Feher da Silva & Hare, 2018), and to test if this affects the observed associations with individual differences in a manner we should be concerned with. As described in our paper, we found some early evidence that it matters - mean-level model-based estimates were larger when estimated using Drift B, but the association between model-based estimates and compulsivity was stronger when estimated using Drift A.

REFEREES' REVISIONS & AUTHORS' RESPONSES FOR COMMSPSYCHOL-23-0053-T

Given this finding, we aim to analyse data from the other 8 drift sets we implemented in the free-play section of the app and identify candidate features of drifts with better or worse signal, in a data-driven way. We then plan to test these predictions in new drifts developed more systematically to have certain features more than others. Unfortunately, at the time of submission, we did not have sufficient sample size to present this work. However, for the interested reader, we have now include the reward trajectories of all ten drifts used in Cannon Blast (Supplementary Material Page 5), along with descriptive statistics (Supplementary Material Page 12). These drifts differ in various properties such as average reward rate, reward distinguishability and reward variability. We included, for the purpose of this response letter only – but not in the paper, the MBI coefficients across these drifts. Given the N is not yet where we want it to be for these analyses, we prefer to wait to examine this in more detail in a future publication where we can also assess individual differences.

Figure S3. Reward trajectories for each of the ten drifting reward probabilities sets. On participants first play ('Risk Factors') participants were randomly assigned to receive either Drift Set A or Drift Set B at each block of trials. On participants repeated plays ('Free Play'), participants were randomly assigned to receive one out of the possible ten drifts (A – J).

Table S6. Descriptive Information of Reward Drift Sets

Reward Drift Set	Mean Reward Probability		Mean Reward Probability	Mean Reward Difference	SD Reward Probability		Mean SD Reward	Mean SD Difference
	Purple	Pink			Purple	Pink		
A	.845	.748	.797	.096	.053	.049	.052	.005
B	.774	.501	.638	.273	.103	.041	.072	.062
C	.649	.829	.739	.180	.113	.088	.101	.025
D	.756	.809	.783	.053	.109	.075	.092	.034
E	.542	.611	.576	.069	.110	.059	.084	.051
F	.649	.647	.648	.002	.043	.096	.070	.053
G	.909	.575	.742	.335	.066	.081	.074	.015
H	.510	.564	.537	.054	.043	.096	.069	.053
I	.849	.698	.774	.151	.062	.092	.077	.030
J	.532	.906	.719	.374	.069	.043	.056	.026

Note: Only Drift Set A & Drift Set B used in participants first play ('Risk Factors' section of the app). All ten drifts (A-J) used in repeated plays of Cannon Blast ('Free Play section of the app)
 Mean reward/SD difference=Absolute value between the highest mean reward/SD probability minus the lowest mean reward/SD probability

REFEREES' REVISIONS & AUTHORS' RESPONSES FOR COMMSPSYCHOL-23-0053-T

Results: Model-based coefficients across all drifts sets used in app. We will revisit future work once a sufficient amount of data is collected per drift set.

We have reframed our conclusions in the text (Page 14, Line 523-527) to address concerns regarding the generalisability of our findings related to drift rates. Instead, we have discussed the potential for future work involving the exploration of the other drift rates once we have sufficient power to do so and as suggested complementing this research by systematically varying the random walks.

“One suggestion is to revisit these analyses using all ten drift sets available in the app, once we have collected a sufficient amount of data per drift. Any candidate drift properties (e.g. boundaries, variance, drift rate) that affect the relationship between model-based planning and compulsivity can later be more systematically confirmed in new drift sequences optimised to maximally compare relevant dimensions.”

REVIEWER #2

"Donegan and colleagues report their findings of two experiments that employ a novel, gamified, version of the two-step task, which measures the degree to which people engage in model-based vs model-free reinforcement learning. They show that the task has reasonable cross task reliability with a more conventional version of the task. Moreover, they show that the new task has a few advantages over the original, with the most noteworthy insight being that reliable estimates can be obtained with relatively few trials. In addition, they claim to replicate a host of findings, including that concurrent task demands and changes to task structure can decrease model-based control, and that several individual differences (age, sex, IQ) predict use of model-based control.

I think this is an interesting paper, with a cool, inventive, and novel way to study model-based decision making. There is a large potential for this version of the task to transform the way people study model-based reinforcement learning. However, at the same time, I have several concerns and questions that would need to be addressed before this paper is ready for publication."

We would like to extend thanks for your supportive feedback and constructive comments. We have addressed all the following concerns below which we believe has significantly improved the strength of our manuscript.

"First, I am not particularly convinced about the difficulty/task-order analyses. If I understand correctly, the authors are trying to use these to replicate Otto et al. (2013), who showed that maintaining information in working memory during decision making reduced model-based control. It's not clear to me that changes in difficulty here constitute the same manipulation. For example, if I understand the supplied videos, all of these difficulty manipulations are about making the target harder to hit. Therefore, the participant can first decide which "box" they will choose (based on recent information), and then start aiming with the cannon. Note that in this scenario, all model-based planning will have already occurred before aiming difficulty can influence the decision. A more direct replication would have introduced demands that irrefutably would be present throughout the trial."

Thank you for raising the concerns regarding the difficulty/task-order analyses in our study. We understand that you are unsure if these analyses effectively replicate Otto et al.'s (2013) methods and findings on the impact of working memory on model-based control. We agree that in this task, the model-based update would have already occurred by the time someone aims and that it is a much weaker intervention than that used by Otto et al., 2013. Our rationale was that by limiting the availability of cognitive resources through the challenging aiming and timing task, we would tax the working memory resources required for the *retention* of information necessary for model-based planning. The results however, fall in agreement with the reviewer – that the manipulation was simply not strong enough.

We have added information about what we hypothesised in terms of how difficulty would impact behaviour (Page 9, Line 334-339).

"Difficulty in Cannon Blast referred to how challenging the diamond shooting task was on a given trial, with harder levels involving more timing and spatial reasoning. We hypothesised that this would put a strain on the cognitive resources required for maintaining an accurate model of the task environment and therefore impede model-based planning."

We appreciate your feedback and understand the importance of a more direct replication by introducing demands that are consistently present throughout the trial which we have added in the discussion as a limitation to the interpretation of the results regarding difficulty (Page 13, Line 486-493). We will carefully consider this suggestion for future research.

REFEREES' REVISIONS & AUTHORS' RESPONSES FOR COMMSPSYCHOL-23-0053-T

“We did not find the association between model-based planning and compulsivity significantly differed between manipulations made to task difficulty. However, this could be a limitation of our task design, which used a relatively weak dual task manipulation. Specifically, difficulty in this context referred to the extra demands placed on participants to shoot the diamond on harder levels, which required taking into account the geometry of the screen and timing the shot. Perhaps crucially, this demand occurs after the model-based update had occurred and participants have presumably already decided which container to fire from. Future research would require a more potent dual-task manipulation to assess the impact of working memory demands on model-based planning in this task (Otto et al., 2013).”

*“Second, and related to this, the task introduces a novel skill-related component, and the authors largely gloss over the implications of this in their task. What’s striking to me about this is that participants only hit targets on 36.5% of all shots on which they received a good ball. This greatly reduces the efficacy of any of the RL systems — their outputs typically do not lead to a win. In fact, it looks like participant only score on ~27% of trials (75%*36.5%). Isn’t it possible that this induces the reduction in model-based control over time? That is, with such a low hit rate, and with computational demands for the MB system, participants learn that it’s not worth it to plan the next choice of ball: it will like lead to a loss anyway. Another unexplored facet of this additional feature in the task is that there are now essentially two forms of feedback: (1) whether or not a “good” ball was drawn, and (2) whether or not a diamond was scored. Does this second reward signal contribute to subsequent choice? It’s easy to see how a model-free system would update actions values according to this second reward signal. Apart from the section on “broader patterns” (which mainly reports correlations), I did not see an in-depth analysis on this feature of the task.”*

You correctly highlight that participants in our study only hit the target on ~27% of the trials. We take on board your idea that if participants experience a low hit rate, they might feel engaging in MB control is not be worthwhile. However, when we look at the comparison of the same participants playing Easy vs Medium levels on their second play (having removed the order confound associate with the first block), participant have different hit rates (Easy: M=25, SD=10.7, Medium: M=18, SD=9.37, $t(334)=6.90$, $p<.001$), but no significant difference in MB planning (Easy: M=.22, SD=.23, Medium: M=.25, SD=.32, $t(334)=-.03$, $p=.081$) and if anything go in the opposite direction to suggested i.e., greater diamonds hit in Easy but nominally smaller model-based planning. This again demonstrates that in this task our difficulty manipulation was too weak which we have highlighted throughout in text and the above point.

Regarding the presence of two forms of feedback in the task. We agree that understanding the impact of this second reward signal (hitting the diamond) is important to flesh out. In response to Reviewer 1, we tested the impact of this secondary reward signal on behaviour by adding it as an additional variable into the model along with reward and transition. Contrary to some intuitions, the effect of diamond hit was to in general suppress learning – as we outline above and in the quoted text below. We believe this is due to the interruption that occurs following a diamond hit – in that the diamond moves location on those trials. We have added these results into the manuscript (Page 8, Line 286-298).

“This would also mean that on ~27% of all trials, users received not only a ‘good ball’, but also received the additional reward of hitting a diamond, which increased their score. This additional reward could plausibly impact model-based/free behaviour by amplifying the reward signal on those trials. Another possibility however is that because the diamond moves location after it is successfully hit, this could in fact interrupt learning and so have the opposite effect. We tested this by entering ‘Diamond Hit’ into the model with reward and transition on stay behaviour. We found a negative main effect of lagged diamond hit on stay behaviour, though this failed to reach statistical significance ($\beta=-.05$ (.03), $p=.053$). Similarly, on trials following a diamond hit, users had reduced model-free

REFEREES' REVISIONS & AUTHORS' RESPONSES FOR COMMSPSYCHOL-23-0053-T

behaviour (Reward*Diamond Hit interaction, $\beta=-.14(.03), p<.011$), and reduced model-based behaviour (Reward*Transition* Diamond Hit interaction, $\beta=-.09 (.03), p<.011$). Together these findings suggest that the receipt of a diamond functioned to impair learning, not potentiate it."

Table S10. Linear Regressions Predicting Stay Behaviour by Reward, Transition and Diamond Hit on the Previous Trial

	B (SE)	z	p
Intercept	1.38 (.03)	49.33	<.001
Reward	.43 (.03)	16.22	<.001
Transition	.01 (.03)	.42	.676
Diamond Hit	-.05 (.03)	-1.94	.053
Reward * Transition	.19 (.03)	7.22	<.001
Reward * Diamond Hit	-.14 (.03)	-5.39	<.001
Transition * Diamond Hit	.08 (.03)	3.10	.002
Reward * Transition * Diamond Hit	-.09 (.03)	-3.35	<.001

Variable coded as: Reward; No Reward, 1; -1, Common; Rare, 1; -1, Hit; Miss, 1; -1

"My third worry revolves around two recent studies by Feher - Da Silva and Hare in NHB (2020, 2023), that seem to demonstrate that signatures of model-free control are actually better understood as people applying model-based control over faulty models. To be honest, I am skeptical of those results, and I think that the current results could deliver evidence against this interpretation. That being said, I think the authors could do a more thorough job in describing how their design mitigates these concerns."

First, we must state that our task was not designed to address these papers, being designed and released prior to their being published. Nonetheless, we have of course followed these papers with interest.

With respect to model-free learning, one the one hand, the position we hold is similar to these authors - model-free estimates from the classic task are ambiguous in interpretation. For example, our prior work (Gillan et al., 2015) illustrated that model-free signatures on the classic task were not linked to habit learning assessed through devaluation. So the theory supporting them has fallen at some foundation hurdles in our opinion. Feher da Silva and Hare (2020) illustrate nicely that added instructions increase model-based and decrease model-free estimates – but we are not certain that the correct interpretation of this is that the effect of reward on stay behaviour is explained by their use of an incorrect model. Rather that humans under heavy instruction can follow those instructions pretty well.

As the reviewer alludes to, our task may speak to some of the issues raised in these papers around the role of task instruction. A major difference between our task and previously published ones is that we actually display the transition structures on screen throughout the entire task - so there is no learning component at all with respect to action-state transitions. We nonetheless observe signatures of model-free behaviour. However, we are cautious to interpret these too heavily, given (i) the unique set up of our task where there is a second-order task within a task and (ii) model-free estimates in our task may measure something qualitatively different to other task, indeed, they were related to the same individual difference measures as model-based planning estimates were – in contrast to a vast pre-existing literature on the twostep task showing model-free effects have few robust individual difference correlations. For us, the focus was on model-based planning and we are satisfied that despite showing the transition structure

REFEREES' REVISIONS & AUTHORS' RESPONSES FOR COMMSPSYCHOL-23-0053-T

through the game, we were able to elicit useful individual difference measures for MB planning. We now elaborate a bit more on our interpretation of the model-free effects on this task in the discussion (Page 15, Line 572-581):

In the present study, we embedded the classic two-step task within a game, that is, we made model-free and model-based behaviour operate in the service of a reward (gaining good balls), which was itself only valuable insofar as it served the higher-order goal of shooting diamonds. This meant that model-free behaviour was no longer the lowest level of engagement with the task and it is therefore possible that those with the lowest cognitive capacity may have focused exclusively on the primary task of shooting the diamonds, adding new 'model-based' signal to the model-free estimate. Indeed, the cross-task correlations in the present study support this idea as model-free measures from Cannon blast were positively associated to both model-free and model-based estimates measured from the traditional task.

"Fourth, it appears that for all drifting reward probabilities, the purple option is almost always better. This strikes me as odd, isn't the idea of the drifting probabilities that it should keep people on their toes about which option is better? Right now, it seems like people could have just clued in too that the purple is the superior one, and to let this guide choice (potentially reducing the MB influence). Why did the authors not simply implement truly randomly drifting rewards. The authors claim that the typical two-step task only includes one drift sequence, but many modern papers let rewards randomly over time for each participant (one of many papers that do this: Bolenz et al., 2022, Scientific Reports)."

We did not implement random and unique drifts per person, because we wanted to ensure a sufficient number of cases per drift to test the impact of using different drifts on (i) MB planning and (ii) the association between MB planning and compulsivity. We agree that it is a limitation that purple was the best choices for participants in the main game, though of course, they did not know this as they proceeded trial by trial. Of course, while categorically purple was best most of the time for the main drifts in the paper, the difference between the two ranged from trivial to substantial.

To expand on this issue though, in the free-play section, we included 8 additional drifts, which included cases where the most rewarding colour changed during the session (e.g., D and F) and where pink was the most consistently rewarding (e.g., C and J). For the interested reader, we have now included the reward trajectories of all ten drifts used in Cannon Blast (Supplementary Material Page 5), along with reward-related metrics (Supplementary Material Page 12). These drifts differ in various properties such as average reward rate, reward distinguishability and reward variability. We include, for the purpose of this response letter only, the MBI coefficients for these drifts. The drifts that switch 'the best ball colour' midway through look similar to the others. We do see however that the drift that decisively favours pink appears to elicit greater MBI than the others, which could reflect the fact that it is a switch from the first 200 trials where purple was best. Although suggestive, given the N is not yet where we want it to be for these analyses of the 8 drifts, we prefer to wait to publish this in more detail in a future and more focused publication.

In the interim, we have reframed our conclusions in the text (Page 14, Line 523-527) to address concerns regarding the generalisability of our findings related to drift rates. Instead, we discuss the potential for future work involving the exploration of the other drift rates once we have sufficient power to do so and as suggested complimenting this research by systematically varying the random walks.

"One suggestion is to revisit these analyses using all ten drift sets available in the app, once we have collected a sufficient amount of data per drift. Any candidate drift properties (e.g. boundaries, variance, drift rate) that affect the relationship between model-based planning and compulsivity can later be more systematically confirmed in new drift sequences optimised to maximally compare relevant dimensions."

REFEREES' REVISIONS & AUTHORS' RESPONSES FOR COMMSPSYCHOL-23-0053-T

Figure S3. Reward trajectories for each of the ten drifting reward probabilities sets. On participants first play ('Risk Factors') participants were randomly assigned to receive either Drift Set A or Drift Set B at each block of trials. On participants repeated plays ('Free Play'), participants were randomly assigned to receive one out of the possible ten drifts (A – J).

Table S6. Descriptive Information of Reward Drift Sets

Reward Drift Set	Mean Reward Probability		Mean Reward Probability	Mean Reward Difference	SD Reward Probability		Mean SD Reward	Mean SD Difference
	Purple	Pink			Purple	Pink		
A	.845	.748	.797	.096	.053	.049	.052	.005
B	.774	.501	.638	.273	.103	.041	.072	.062
C	.649	.829	.739	.180	.113	.088	.101	.025
D	.756	.809	.783	.053	.109	.075	.092	.034
E	.542	.611	.576	.069	.110	.059	.084	.051
F	.649	.647	.648	.002	.043	.096	.070	.053
G	.909	.575	.742	.335	.066	.081	.074	.015
H	.510	.564	.537	.054	.043	.096	.069	.053
I	.849	.698	.774	.151	.062	.092	.077	.030
J	.532	.906	.719	.374	.069	.043	.056	.026

Note: Only Drift Set A & Drift Set B used in participants first play ('Risk Factors' section of the app). All ten drifts (A-J) used in repeated plays of Cannon Blast ('Free Play section of the app)

Mean reward/SD difference=Absolute value between the highest mean reward/SD probability minus the lowest mean reward/SD probability

Results: Model-based coefficients across all drifts sets used in app. We will revisit future work once a sufficient amount of data is collected per drift set.

REFEREES' REVISIONS & AUTHORS' RESPONSES FOR COMMSPSYCHOL-23-0053-T

"I found the section on difficulty and task-order hard to read. I think this is partly because these two components are confounded, but also because it never becomes exactly clear what the authors conclude from the findings. This is partly because it's unclear what the "free play" session entails, and how these involve "first" and "second" blocks. Moreover, it is unclear what conclusions the authors draw from the analyses of data in this phase? Do they think task difficulty decreases model-based control? Do they think task order influences model-based control? Or are these components so confounded that it's truly hard to determine what can be concluded. I think this section would benefit from a more explicit description of the conclusions."

This was also raised by another reviewer. To address the comments: First, we have made clear what our predictions were with respect to difficulty (Page 9, Line 334-339).

"Difficulty and Order Effects. Difficulty in *Cannon Blast* referred to how challenging the diamond shooting task was on a given trial, with harder levels requiring more timing and spatial reasoning to hit the target than easy trials (where the diamond was unobstructed or static). We hypothesised that this would put a strain on the cognitive resources required for maintaining an accurate model of the task environment and therefore impede model-based planning."

Next, have improved our description of the distinction between 'Risk Factors' (participants initial play of the game, two runs of 100 trials) and 'Free Play' (participants subsequent plays, 100 trials) (Page 9:11, Line 339-381).

"As part of their first play of *Cannon Blast* during the 'Risk Factors' section, all participants completed two blocks of trials; a block of 100 Easy difficulty trials (*'Easy-1st Block'*), followed by a block of 100 Medium difficulty trials (*'Medium-2nd Block'*). In each of the two blocks (*Easy-1st Block*, *Medium-2nd Block*), we observed main effects of Reward, Transition, and their interaction (**Table S7**). However, we found MBI estimates were significantly larger in the Easy-1st (M=.30, SD=.32) than Medium-2nd (M=.22, SD=.29) blocks, $t(5004)=20.16$, $p<.001$ (**Table 2, Figure 4A(ii)**). Because difficulty and order are confounded with one another, we cannot confirm if this reduction was driven by increased task difficulty, order, or both. To isolate the effect of difficulty from order, we turned to data from the 'Free Play' section of the app, where a subset of participants re-engaged with *Cannon Blast* at a chosen difficulty level for short games of 100 trials. We compared their initial play ('first play') at each difficulty level (Easy, Medium) with their next play ('second play') of that difficulty. In N=689 participants who had completed two sessions of *Cannon Blast* at an Easy difficulty, we found MBI was greater at their first play (M=.32, SD=.33) compared to their second (M=.21, SD=.23), $t(688)=8.91$, $p<.001$. Importantly, we did not find this difference when we repeated this analysis in N=556 who had two sessions of play at Medium difficulty [first play (M=.24, SD=.33); second play (M=.25, SD=.33), $t(555)=-.59$, $p=.552$]. This suggests that our difficulty manipulation did not affect model-based planning, but that there was an effect of order – MBI estimates are greatest during the first play of *Cannon Blast*, which reduces and stabilises across subsequent sessions. To confirm this, we compared model-based estimates in N=335 who had sessions at both difficulties (*Easy*, *Medium*) and at two time points (*First play*, *Second play*). We found MBI at *Easy-First play* (M=.33, SD=.35) was significantly larger than all other times, including *Medium-First play* (M=.25, SD=.34, $p=.002$), *Easy-Second play* (M=.22, SD=.23, $p<.001$), and *Medium-Second play* (M=.25, SD=.32, $p=.002$). There were no other significant differences (all $p>.6$). More evidence for the uniqueness of the first block in which people play comes from an analysis of test re-test reliability of task sections. MBI estimated from the Easy-1st Block of the game was weakly associated with their second sessions at Easy (N=689, $ICC_1=.35$). The same analysis of Medium plays yield higher reliability (N=556, $ICC_1=.51$) (**Table S13**). In terms of reliability within each block, we found comparable split-half reliability estimates (Easy-1st Block: $r=.72$ [95%CI =.70-.74]; Medium-2nd Block $r=.68$ [95%CI =.66-.70], **Table 2**.)"

We have also added detail into our conclusions (Page 13, Line 486-493):

REFEREES' REVISIONS & AUTHORS' RESPONSES FOR COMMSPSYCHOL-23-0053-T

“We did not find the association between model-based planning and compulsivity significantly differed between manipulations made to task difficulty. However, this could be a limitation of our task design, which used a relatively weak dual task manipulation. Specifically, difficulty in this context referred to the extra demands placed on participants to shoot the diamond on harder levels, which required taking into account the geometry of the screen and carefully timing the shot. Perhaps crucially, this demand occurs after the model-based update had occurred and participants have presumably already decided which container to fire from. A more potent dual-task demand would impact the update itself (Otto et al., 2013).”

“If I understand correctly, “bad balls” are observed for a shorter period of time than “good balls”. This appears to change demands for wins and losses (e.g., outcomes need to be processed faster than losses). Have the authors considered the effect of this on their task? For example, during computational modeling, they could include separate learning rates for wins and losses. This is already an established practice for RL tasks, but here it seems particularly important to account for this change in the paradigm.”

Thank you for highlighting this important feature of the task regarding the duration of rewarding compared with non-rewarding trials. You are correct that, on average, rewarding trials (‘good ball’) are observed for longer than non-rewarding trials (‘explosion’ for 1000ms). We added an additional model (model G) to the supplement that expands on Model D with separate learning rates for positive vs negative feedback (Supplementary Material Page 29). While it converged and performed well across most metrics, it had a larger LOOIC compared to model D and so we retain the simpler model.

Table S18. Candidate Model Comparison and Evaluation for Hierarchical Bayesian Models

Model Name	A	B	C	D	E	F	G
MCMC diagnostics							
Divergence	78	371	0	0	1	2	0
All rhats <1.05	NO	YES	YES	YES	YES	YES	YES
LOOIC	17283	16637	17310	16486	16495	16565	16533
PPC							
Mean % agree	0.81	0.82	0.81	0.82	0.82	0.82	0.82
Median % agree	0.82	0.84	0.82	0.84	0.84	0.84	0.84
Prop. over 80% agree	0.59	0.66	0.59	0.63	0.64	0.64	0.60
Min	0.59	0.53	0.60	0.60	0.58	0.59	0.64
Max	0.96	0.96	0.96	0.96	0.96	0.96	0.97
Parameter Recovery							
ρ Perseveration	0.96	0.96	0.96	0.97	0.97	0.97	0.97
α_1 1st stage LR	0.73	0.82	0.79	0.87	0.86	0.79	-
α_2 2nd stage LR	0.64	-	-	-	-	-	-
β_1 Inverse temp.	0.74	-	-	-	-	-	-
ω MB/MF weight	0.63	-	-	-	-	-	-
β_{MB} Model-based β	-	0.74	0.70	0.71	0.72	0.74	0.73
β_{MF} Model-free β	-	0.78	0.65	0.75	0.78	0.82	0.81
α_D Decay rate	-	-	-	-	0.35	-	-
α_+ 1st stage LR+	-	-	-	-	-	-	0.75
α_- 1st stage LR-	-	-	-	-	-	-	0.79

All models were run on n=100 subjects (*simulated agents* for parameter recovery), with models estimated from 4K iterations. For each row, the best performing model (or joint-best) is highlighted in green.

REFEREES' REVISIONS & AUTHORS' RESPONSES FOR COMMSPSYCHOL-23-0053-T

“For the trial number analysis, it is not clear whether the graph shows that simply increasing the number of trials entered in to the model-fitting procedure reduces the estimate the model-based component, or whether MB control shows a reduction over time. That is, for the data points labeled “25 trials”, are these the first twenty five trials in the task, or just 25 randomly picked trials? Either of these options have considerable caveats. If the former, then the reduction in MB control over trials is not surprising, because later trials involved more difficult task aspects. If the latter, then the conclusions drawn from this analysis, that you can get robust estimates with few trials, is a bit misleading because this scenario is not the same as what would happen if one only runs 25-trial two-step tasks (where the trials are consecutive). This section requires clarification, and the authors should hedge the conclusions they draw from this section.”

Thank you for highlighting this, which has allowed us to add more clarification to the interpretation of our trial number analyses and presented graph. The data point labelled ‘25 trials’ represents participants’ first 25 trials (1-25) in the task, followed by data points for the first 50 trials (1-50), and so on, cumulatively calculated. We apologise for the lack of clarity in the description of this and have revised the graph and its description (Page 11, Line 418-421) text to explicitly state how these data points represent participants cumulative trials in order of presentation.

“To test the impact of increasing the total number of trials used to estimate MBI per participant, we generated each participant’s MBI several times, starting with a participant’s first 25 trials and increasing by 25 trials in each iteration, until 300 trials per participant.”

You correctly highlight there is a caveat to the interpretation of model-based estimates reducing with the addition of more trials. We agree that this finding is non-surprising given that we showed in our manuscript model-based estimates decline from the 1st Block/Easy trials to 2nd Block/Medium trials. However, as per analysis reported above, we also show that difficulty itself does not greatly impact model-based planning i.e., model-based estimates were significantly greater at first play than follow-up up at the same difficulty level, and no differences between model-based estimates from a follow-up Easy block with first or follow-up blocks at a Medium difficulty. This would suggest our manipulation of task difficulty was perhaps too weak but also that there is a certain uniqueness about a participants first play in Cannon Blast. What is more interesting, and a key finding from these analyses is the association between model-based planning and CIT can be estimated using just participants’ first 25 trials.

We have used this opportunity to expand more on these analyses, and have added a complementary analysis to this section of the manuscript looking at the specific bins of 25 trials (1-25, 26-50, 51-75 etc) and comparing to another publicly available dataset using the traditional version of the task to complement this analysis (Page 13, Line 457-469). As is evident in the plots below, the first trials of our task have more signal for individual differences (Panel Aii– which is in stark contrast to the traditional task (Panel Dii):

“Next, we wanted to test if any specific collection of trials were driving these effects. To do this, we repeated the above analyses using bins of 25 trials (0-25, 26-50, etc). Similar to cumulative trials, we found an overall reduction in MBI for later trial bins (**Figure 5Aii**). Reliability estimates were relatively stable across the bins of trials (**Figure 5Bii**). But associations between MB and CIT reduce as the task progresses (stat, **Figure 5Cii**). To test if this pattern is specific to our task, we carried out the same analysis on data from the traditional task in a previously published study (N=1413 who completed the traditional two-step task remotely online)(Gillan et al., 2016). In the traditional task, participants must learn the transition probabilities through trial and error (unlike our task where they are shown on-screen throughout), so it follows that the early trials should not be informative for assessing model-based planning. Indeed this was the case; in the traditional task, associations with CIT increase with more trials when examined cumulatively (**Figure 5Di**) and associations with CIT were absent from the first 2 bins of trials, and only become significant later (**Figure 5Dii**). ”

REFEREES' REVISIONS & AUTHORS' RESPONSES FOR COMMSPSYCHOL-23-0053-T

Figure 5. The impact of increasing number of trials on mean-level model-based estimates, its reliability and their association with individual differences (N=716). **A.** Mean and standard error of MBI estimated with (i) cumulative trials i.e., participants' first 25 trials increasing sequentially in bins of 25 trials until 300 trials per participant was reached, and (ii) binned trials i.e., bins of 25 trials. Mean-level MBI decreased with more trials. **B.** Split-half reliability co-efficient with 95% confidence intervals as a function of increasing trial number using (i) cumulative trials and (ii) binned trials. Using the cumulative trials, reliability of the MB estimate increased with additional trial collected per participant. **C.** Model-based associations with compulsivity, age, gender and education using (i) cumulative trials and (ii) binned trials. Increasing the number of trials per participant did not increase the association between model-based planning and individual differences. No significant difference was found between model-based associations with compulsivity when estimated at participants' first 25 vs 300 trials. Model-based associations with age and education became stronger with the addition of trials while the association between model-based and gender reduced. Data from the binned analysis demonstrated that this effect is in part driven by stronger signal in earlier versus later trials. **D.** We repeated this analysis using a publicly available dataset of N=1413 individuals who completed the traditional two-step task (200 trials). Here we found (i) the association between model-based planning and individual differences increased as trials collected per participant increased and (ii) this effect is driven by later trials compared to earlier trials.

We have added more discussion of this in the results summary that appears at the end of the introduction (Page 3, Line 126-131):

“This paradoxical finding was explained by the fact that early trials on our task carried the greatest signal for compulsivity. By analysing a publicly available dataset, we show the situation is reversed for the traditional task; Early trials carry the least signal, presumably because transition probabilities are not yet learned, while in our task, they are displayed explicitly on-screen.”

REFEREES' REVISIONS & AUTHORS' RESPONSES FOR COMMSPSYCHOL-23-0053-T

And in the discussion (Page 14, Line 552-558):

This early signal in the task stands in contrast to the traditional version, where we show that early trials do not carry much individual difference signal. We speculate that this difference is because in the traditional task, participants must learn the transition probabilities through trial and error (and so model-based planning cannot be executed from trial 1), whereas in Cannon Blast, the probabilities are displayed on-screen. This simple change in design may have major implications for future studies aiming to reduce the trials required to estimate model-based planning.

"I am genuinely confused why the participants in the second phase are called "citizen scientists". If I understand correctly, these people just took part in the task, but did conduct any of the research? Maybe this is a term that the particular app developed to entice people to take part in their studies, but I don't think it fits a scientific publication."

The term citizen scientist refers to "the *collection* and analysis of data relating to the natural world by members of the general public, typically as part of a collaborative project with professional scientists." (Oxford Languages). In the case of our study, participants do not analyse data, but they *collect it* by engaging in introspection and entering data on their thoughts feelings and experiences into an app, much like how citizen scientists in botany do so when recording their observations of the natural world.

The core component of citizen science is not about analysis necessarily, but using crowdsourcing and people-power to facilitate large research projects not otherwise possible or practical. From a recent paper: "The broad global reach of the internet has opened up a superhighway of opportunity to people previously excluded from academic science and allowed us to design ways to take large-scale science challenges to the global marketplace instead of hoping they will find us. This growing field—citizen science—engages people from beyond the traditional academic arena either to collect and generate data or to contribute as individuals or teams to analyze it." We have added this citation (Page 3, Line 107-109) to clarify the use of this term in our paper.

"This requires large samples, and so we developed a novel diamond-shooting game called *Cannon Blast* that could be played by members of the public, aka 'citizen scientists' (Roskams & Popović, 2016), from anywhere in the world in an at-home environment."

"The authors mention "model-free planning" (page 8), and that seems like a contradiction in terminis."

Thanks for your attention to detail in identifying this typographical error, this has been rectified (Page 8, Line 262)

"However, individual differences in model-free learning behaved similarly to model-based planning in our task."

REVIEWER #3

"I was asked to comment on data protection issues and ethical issues regarding the smartphone-based approach used in this work – but could not find sufficient information in the paper. Given the importance of these issues, particularly regarding the fact that sensitive (mental-health-related) data are collected, I consider it important that the authors describe data protection procedures, terms of data use, and what participants gave their consent for in detail. Also, information of whether there was an ethics committee approval of the current study is essential. I found some statement that "Neureka has received ethical approval from the School of Psychology Committee on Research Ethics and is fully GDPR compliant." on the internet, but it should be made clear in the paper in which way the present work is covered by this general approval.

Otherwise, I think that this an important and methodologically sound study. Methods are well described, results clearly presented, and conclusions well supported. Particularly the results on the impact of increasing the number of trials on parameter estimates, their reliability, and associations with other variables are highly interesting and generally relevant for researchers who want to create online versions of certain tasks and (could) have them done extensively by participants.

I just have a few minor comments, primarily on further information that could be provided."

We would like to thank you for your encouragement of this work and for the insights into data protection and ethical considerations for this approach. We acknowledge the importance of addressing these issues, especially given the sensitive nature of the mental health-related data collected. We apologize for the lack of information provided in the paper regarding data protection procedures, terms of data use, and participant consent details. We agree that these aspects should be clearly described, and a dedicated section addressing all of these concerns has been added to the paper ('Ethical Considerations and Data Protection', Page 21, Line 852-861).

"This research was granted ethical approval by Trinity College Ethics Committee (Approval number: SPREC072019-01). For Experiment 1, prospective participants received an information sheet and gave informed consent through the online survey platform Qualtrics. Participant across both experiments were required to also read the information sheet and consent to participation embedded to the registration process for the Neureka app. This described the wider scientific aims of the Neureka Project, what participation involves, terms of data use, data protection procedures, health risks, withdrawal of data procedures and points of contact. For more detail on the exact contexts of this information sheet provided to participants, see Supplementary Material Note 1. Data collected through Neureka is stored and processed in accordance to EU General Data Protection Regulations."

"It may be interesting to get some more information on how the difficulty of the shooting task was set. Was this somehow calibrated so to be maximally motivating for participants? The authors write that "the task was fairly challenging", but it is not clear whether this was intended and whether it may be too difficult for certain groups (particularly older adults)."

We appreciate your comments regarding the difficulty of the task. We conducted internal testing to calibrate the difficulty levels and ensure it was engaging and motivating but also challenging. We did consider the potential variability in individual differences particularly for some groups like the elderly and so conducted testing with members from the general public. The task's difficulty was designed to strike a balance between providing a stimulating fun experience and avoiding it being excessively challenging. We compared diamond hit rates across levels to confirm the manipulation achieved a tiered difficulty level which we have now added in text (Page 18, Line 669-672):

REFEREES' REVISIONS & AUTHORS' RESPONSES FOR COMMSPSYCHOL-23-0053-T

“While on average medium trials are more difficult than easy (average hit rate Medium=45%, Easy=52%), there was variation within both Easy levels (hit rates 83%, 53%, 44%, 29%) and Medium (hit rates 75%, 20%, 39%, 45%).”

We also found older adults showed reductions in model-based planning and diamond hit rates (controlled for good balls received), Page 8 Line 276.

Table S2. Descriptive Information of Easy and Medium Levels of Difficulty in Cannon Blast

Difficulty	Level	Description	Diamond Behaviour	Hit Rates
Easy	1	Static diamond		83%
Easy	2	Moving diamond horizontally or vertically		53%
Easy	3	Moving diamond in direction of an arc		44%
Easy	4	Static diamond facing down in 3 directions		29%
Medium	5	Static diamond with shell facing down		75%
Medium	6	Moving diamond diagonally with shell facing down		20%
Medium	7	Static diamonds with rotating shells		38%
Medium	8	Static diamonds with gates		45%

N = 57 in Exp 1 (targeted sample size was a minimum of 50) may be sufficient to get medium-sized effects significant but is not a large sample to estimate the correlation of the different task versions with high precision. What is the CI of the reported correlation of .40?

Thank you for this observation. Regarding the reported correlation, the confidence interval for this correlation was [.16, .60], which suggests a moderate to strong positive relationship between model-based planning estimated from the traditional and Cannon Blast in a sample of N=57.

I would recommend using one alpha level throughout (i.e., refrain from using different “significance levels”).

We have edited the manuscript in line with the journals policy on reporting p-values. That is p-values must be reported exactly, unless $p < .001$ or for p-values $>$ alpha level, defined as .05.

REFEREES' REVISIONS & AUTHORS' RESPONSES FOR COMMSPSYCHOL-23-0053-T

Regarding the matched samples (reported in Table S7), it would be interesting to get some descriptive information (means and SDs of the 3 matching variables in the two samples) about how well the achieved matching was.

Thank you for highlighting this. Your suggestion has been considered and added to our Supplementary Material (Page 9). From this we see that all three matching variables (age, gender and education) were adequately matched across the groups.

Table S3. Demographic Characteristics in Experiment 2 in the Whole Sample and the two Age, Gender and Education Matched Samples.

Characteristics	Whole Sample	80:20 Matched	70:30 Matched
N	5005	2138	2138
Mean (SD)			
Age	45.69 (14.54)	47.23 (13.62)	48.12 (14.42)
N (% of N)			
Gender			
Cisgender female	3225 (64%)	1397 (65%)	1429 (67%)
Cisgender male	1683 (34%)	701 (33%)	661 (31%)
Non-cisgender ^a	82 (1.9%)	37 (1.9%)	44 (1.9%)
Preferred not to say	15 (>.01%)	3 (>.01%)	4 (>.01%)
Education Attainment			
No formal education	71 (1%)	21 (1%)	25 (1%)
Lower Second Level	399 (8%)	161 (8%)	167 (8%)
Upper Second Level	1314 (26%)	579 (27%)	594 (27%)
University/College	2043 (42%)	846 (39%)	876 (41%)
Master's	958 (19%)	434 (20%)	377 (18%)
PhD (or equivalent)	219 (4%)	97 (5%)	99 (5%)

^a 'Non-cisgender' includes those who identify as Transgender Male, Transgender Female, Non-Binary, or not-listed
SD=Standard deviation

References

- Feher da Silva, C., & Hare, T. A. (2018). A note on the analysis of two-stage task results: How changes in task structure affect what model-free and model-based strategies predict about the effects of reward and transition on the stay probability. *PLoS One*, *13*(4), e0195328. <https://doi.org/10.1371/journal.pone.0195328>
- Feher da Silva, C., & Hare, T. A. (2020). Humans primarily use model-based inference in the two-stage task. *Nature Human Behaviour*, *4*(10), 1053-1066. <https://doi.org/10.1038/s41562-020-0905-y>
- Gillan, C. M., Kosinski, M., Whelan, R., Phelps, E. A., & Daw, N. D. (2016). Characterizing a psychiatric symptom dimension related to deficits in goal-directed control. *Elife*, *5*, e11305. <https://doi.org/10.7554/eLife.11305>
- Gillan, C. M., Otto, A. R., Phelps, E. A., & Daw, N. D. (2015). Model-based learning protects against forming habits. *Cognitive, Affective, & Behavioral Neuroscience*, *15*(3), 523-536. <https://doi.org/10.3758/s13415-015-0347-6>
- Otto, A. R., Gershman, S. J., Markman, A. B., & Daw, N. D. (2013). The curse of planning: dissecting multiple reinforcement-learning systems by taxing the central executive. *Psychological science*, *24*(5), 751-761. <https://doi.org/10.1177/0956797612463080>
- Roskams, J., & Popović, Z. (2016). Power to the people: addressing big data challenges in neuroscience by creating a new cadre of citizen neuroscientists. *Neuron*, *92*(3), 658-664.

21st Aug 23

Dear Claire,

Your manuscript titled "Using smartphones to optimise and scale-up the assessment of model-based planning" has now been seen by our reviewers, whose comments appear below. In light of their advice I am delighted to say that we are happy, in principle, to publish a suitably revised version in *Communications Psychology* under the open access CC BY license (Creative Commons Attribution v4.0 International License).

We therefore invite you to revise your paper one last time to address a list of editorial requests. At the same time we ask that you edit your manuscript to comply with our format requirements and to maximise the accessibility and therefore the impact of your work.

Your Code and Data Availability statements are currently not aligned with our policies. We recognize that the sensitive nature of (some) of the data may stand in the way of public sharing of the full dataset. However, we still require public sharing the numerical data underlying the display items, and the of the code. We also require more details in the Data Availability Statement on what restrictions on access of the full (deidentified) data exist. I have included much more information on this, as well as on other editorial requirements that are currently unmet, in the attached "Editorial Requests Table".

EDITORIAL REQUESTS:

SUBMISSION INFORMATION:

OPEN ACCESS:

Communications Psychology is a fully open access journal. Articles are made freely accessible on publication under a [CC BY](http://creativecommons.org/licenses/by/4.0) license (Creative Commons Attribution 4.0 International License). This license allows maximum dissemination and re-use of open access materials and is preferred by many research funding bodies.

For further information about article processing charges, open access funding, and advice and

support from Nature Research, please visit https://www.nature.com/commpsychol/article-processing-charges

At acceptance, you will be provided with instructions for completing this CC BY license on behalf of all authors. This grants us the necessary permissions to publish your paper. Additionally, you will be asked to declare that all required third party permissions have been obtained, and to provide billing information in order to pay the article-processing charge (APC).

* TRANSPARENT PEER REVIEW: Communications Psychology uses a transparent peer review system. On author request, confidential information and data can be removed from the published reviewer reports and rebuttal letters prior to publication. If you are concerned about the release of confidential data, please let us know specifically what information you would like to have removed. Please note that we cannot incorporate redactions for any other reasons.

[link redacted]

Best wishes,

Marike

Marike Schiffer, PhD
Chief Editor
Communications Psychology

REVIEWERS' COMMENTS:

Reviewer #1 (Remarks to the Author):

The authors have adequately addressed my concerns; I have no further comments.

Reviewer #2 (Remarks to the Author):

I have no further comments. I congratulate the authors on very thorough and convincing revision, and on writing a great paper!

Reviewer #3 (Remarks to the Author):

The authors have responded thoroughly and convincingly to my earlier comments and questions, so that I'm happy to recommend acceptance of the article.

The CI of the correlation of the different task versions should also be reported in the manuscript (and not only th rebuttal letter).